# Taming the Long Tail in Human Mobility Prediction

**Xiaohang Xu[1], Renhe Jiang[1]\*, Chuang Yang[1], Zipei Fan[1], Kaoru Sezaki[1]**
[1]The University of Tokyo
xhxu@g.ecc.u-tokyo.ac.jp
{jiangrh, chuang.yang}@csis.u-tokyo.ac.jp
{fanzipei, sezaki}@iis.u-tokyo.ac.jp

## Abstract

With the popularity of location-based services, human mobility prediction plays a key role in enhancing personalized navigation, optimizing recommendation systems, and facilitating urban mobility and planning. This involves predicting a user's next POI (point-of-interest) visit using their past visit history. However, the uneven distribution of visitations over time and space, namely the long-tail problem in spatial distribution, makes it difficult for AI models to predict those POIs that are less visited by humans. In light of this issue, we propose the **Lo**ng-**T**ail Adjusted **Next** POI Prediction (LoTNext) framework for mobility prediction, combining a Long-Tailed Graph Adjustment module to reduce the impact of the long-tailed nodes in the user-POI interaction graph and a novel Long-Tailed Loss Adjustment module to adjust loss by logit score and sample weight adjustment strategy. Also, we employ the auxiliary prediction task to enhance generalization and accuracy. Our experiments with two real-world trajectory datasets demonstrate that LoTNext significantly surpasses existing state-of-the-art works.

## 1 Introduction

Human mobility prediction is essential in various applications, aiming to forecast the next Point of Interest (POI) a user may visit based on their historical location data, preferences, and patterns [10, 8, 53, 33]. By predicting user movements, it supports urban planning, traffic management, and environmental protection, and provides intelligent personalized Location-Based Social Networking (LBSN) services [7, 37], enhancing people's life quality.

The growth of POI prediction tasks is closely linked to the development of LBSN platforms, where users frequently share their itineraries and reviews, leading to a substantial accumulation of geographical visitation data. However, data collection faces challenges due to network and privacy constraints on mobile devices and the requirement for user authorization to record check-ins. This often results in data being sparse and biased towards popular locations, exhibiting a severe long-tail effect. Currently, these methods fall into two primary categories: *Sequence-based* and *Graph-based* models.

- *Sequence-based* models treat users' trajectories as independent visitation sequences. Existing methods include Recurrent Neural Networks (RNNs) [6, 12, 11], Long Short Term Memory (LSTM) [2, 20, 14, 13] and Gated Recurrent Unit (GRU) [4, 5] for modeling the rich spatial-temporal information implied in the visitation sequence.

- *Graph-based* models focus on building models and data structures to capture trend information in the data to enhance the prediction performance, such as the movement trends among all users [45, 39, 34, 35, 41], geographic adjacency [27, 22, 26], and category transition between POIs [49, 48]. This helps in modeling complex global visitation preferences and the semantic context of locations.

---

*Corresponding author

38th Conference on Neural Information Processing Systems (NeurIPS 2024).

Nevertheless, existing works often overlook the intrinsic long-tailed distribution problem in spatial visitation patterns. As shown in Figure 1, it provides the evidence of long-tailed distribution on Gowalla[1] dataset, a real LBSN dataset. From the visualization results, it is evident that only a few POIs are visited more than 100 times. In addition, the illustrative diagram above the graph presents a hypothetical scenario. The prediction model might inaccurately predict that a user would visit a common location, such as McDonald's (a Head POI), while the user actually visits a less common place like a ramen restaurant (a Long-Tail POI). This highlights the importance of designing models capable of accurately predicting visits to Long-Tail POIs. The concept of a long-tailed distribution, while first extensively studied and addressed within the computer vision (CV) field [44], manifests differently in the context of POI prediction. In POI prediction, long-tailed POIs are embedded within users' complex trajectories. This distinction means that unlike in CV, where long-tailed samples can be selectively augmented to balance datasets, selecting long-tailed POIs without considering the spatial-temporal context in which they occur risks losing crucial trajectory information.

Against this background, to mitigate the long-tail problem in the human next POI prediction task, we propose the **Lo**ng-**T**ail Adjusted **Next** POI prediction (LoTNext)[2] framework, which is a generic framework aimed at optimizing and fully utilizing long-tailed POI information. More specifically, our solution first employs a Long-Tailed Graph Adjustment module to reduce noise and long-tailed nodes in the user-POI interaction graph, thereby mitigating the impact of long-tailed POIs on model performance. Through graph adjustment, the model can more accurately capture spatial-temporal information from trajectory contexts. Furthermore, to pre-

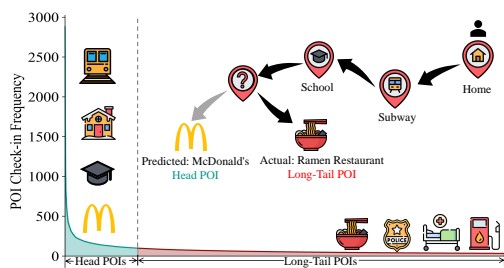

Figure 1: The long-tailed distribution for POI check-in frequency from the Gowalla dataset.

vent the model from overly focusing on head POIs (high-frequency POIs), we propose the Long-Tailed Loss Adjustment module to balance the loss between head and tail POI data. Finally, to alleviate the intrinsic sparsity issue without introducing additional data sources, we incorporate auxiliary prediction tasks to further integrate the POI feature and spatial-temporal information.

We conclude our contributions as follows: (1) We propose the LoTNext framework based on graph adjustment to effectively address the challenges of dataset-inherent sparsity in the user-POI interaction graph. (2) We design the Long-Tailed Loss Adjustment module for adaptive sample re-weighting, which more effectively balances the loss between head and tail samples. (3) We introduce the auxiliary prediction task, which achieves complementarity of POI feature information and spatial-temporal information. (4) We evaluate LoTNext on two public LBSN datasets, comparing it with numerous baselines. The results demonstrate that LoTNext significantly outperforms state-of-the-art methods.

## 2   Related Work

**Next POI Prediction.** Most current works on the next POI prediction treat trajectories as time series, further incorporating spatial-temporal contexts into models to enrich the semantics of POIs. The pioneering ST-RNN [18] introduces spatial-temporal intervals to RNN for context awareness. DeepMove [6] integrates LSTM with attention mechanisms to consider both the short-term and long-term preferences of users comprehensively, and LSTPM [29] enhances spatial context integration. The Flashback model [43] tackles user sparsity by mining similar contexts in historical data. However, due to the limited capability of RNN in modeling long sequences, researchers have explored using graphs for improvements. GETNext [45], based on the Transformer architecture, combines global mobility patterns graph with various spatial-temporal contexts to fully utilize information among similar user trajectories for improving prediction performance. Graph-Flashback [27] considers constructing a knowledge graph to improve POI representation and integrates it with sequence recommendation models. SNPM [46] builds a POI similarity graph to aggregate similar POIs and enhance POI representation results. However, all these studies overlook the significant impact of the long-tail problem on the next POI prediction.

---

[1] https://snap.stanford.edu/data/loc-gowalla.html

[2] https://github.com/Yukayo/LoTNext

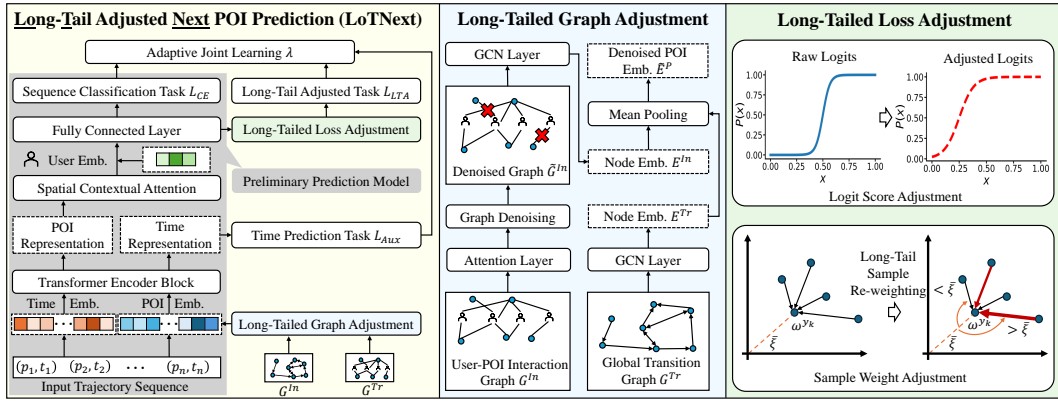

Figure 2: The Architecture of **Lo**ng-**T**ail Adjusted Network for **Next** POI Prediction **(LoTNext)**.

**Long-Tailed Learning.** The long-tail problem has always been a focus in the fields of CV [3, 30] and recommendation systems [15]. The most direct solution is re-sampling [50], varying from random to progressively balanced. Another common strategy is logit adjustment [24, 25, 31], aimed at modifying the logistic output to address the imbalanced data class problem. A study by Google [24] has proven that logit adjustment satisfies Fisher consistency and can effectively minimize the average error per category. Compared to the CV field, the long-tail problem is more pronounced in recommendation systems [1, 19]. Typically, the number of items far exceeds the number of users, leading to many items rarely or infrequently accessed by users. Some works tackle the long-tail problem by learning item similarities through random walk algorithms [47] or utilizing transfer learning to transfer the knowledge from head to tail data [51]. [36] used meta-learning to enhance the information representation of the user-item graph. [21] introduced a novel edge addition module to enrich the connectivity for tail samples. However, unlike traditional recommendation tasks, the next POI prediction involves complex spatial-temporal semantics due to the nature of trajectory data, making it more challenging to improve the representation of long-tailed POI samples. *To the best of our knowledge, our work is the first to propose a general framework for the next POI prediction under the long-tail problem.*

## 3 Problem Definition

Given a user set $U = \{u_1, u_2, ..., u_{|U|}\}$ and a POI set $P = \{p_1, p_2, ..., p_{|P|}\}$, with $|U|$ and $|P|$ indicating the number of users and POIs respectively, we denote the POI check-in as a triplet $\langle u, p, t \rangle$, which means a user $u$ visits POI $p$ at time $t$. Each POI $p$ is a triplet $p = \langle lat, lon, freq \rangle$, representing its latitude, longitude, and visit frequency. We proceed to outline our problem definition as follows.

**Definition 1 (User Next POI Prediction)** *Given a user check-in sequence denoted as $Q_u = (\langle p_1, t_1 \rangle, \langle p_2, t_2 \rangle, \ldots, \langle p_n, t_n \rangle)$, our goal is to predict a list of top POIs that the user $u$ is likely to visit next, which can be taken as a typical sequence classification task over $|P|$ POI candidates. In particular, our work focuses on how to accurately predict the "less visited" POIs belonging to the long-tailed interval.*

## 4 Methodology

In this section, we introduce the details of the LoTNext framework, as shown in Figure 2, which consists of the preliminary POI prediction model, Long-Tailed Graph Adjustment module, and Long-Tailed Loss Adjustment module.

### 4.1 Preliminary Model

We first construct a preliminary end-to-end model, which is designed for precise next POI prediction.

**Trajectory Embedding Layer.** For the embedding generation, we initialize embeddings for POIs $E^P \in \mathbb{R}^{|P| \times d_p}$, timestamps $E^T \in \mathbb{R}^{|T| \times d_t}$, and users $E^U \in \mathbb{R}^{|U| \times d_u}$, where $d_p$, $d_t$ and $d_u$ are the corresponding embedding dimension and $|T|$ is the number of the time slots. In our case, considering each hour slot over a week, there are 168 time slots in total. During the sequence processing phase, we select embedding from $E^P$ and $E^T$ based on the POI and time indices in the input sequence $Q_u$ to construct an embedding sequence $X \in \mathbb{R}^{n \times (d_p + d_t)}$, as $X = \left[ (E_{p_1}^P, E_{p_2}^P, ..., E_{p_n}^P) || (E_{t_1}^T, E_{t_2}^T, ..., E_{t_n}^T) \right]$, where $n$ is the sequence length and $||$ denotes the concatenation operation.

**Transformer Encoder.** Transformer [32] architecture in modeling trajectories has been demonstrated in multiple studies [39, 45, 40], we adopt its encoder block to encode spatial-temporal contexts within trajectories, focusing on capturing long-distance dependencies through multi-layer stacking. To maintain positional information in the sequence, we incorporate a learnable positional embedding $E_{pos} \in \mathbb{R}^{n \times d_p}$ with the raw embedding sequence $X$ to form the Transformer input $\widetilde{X} = X + E_{pos}$. Next, we define the Transformer encoder block as follows:

$$
\begin{aligned}
Z &= \text{LayerNorm}(\widetilde{X} + \text{Multi-Head Attention}(\widetilde{X})), \\
\widetilde{Z} &= \text{LayerNorm}(Z) + \text{FFN}(Z)),
\end{aligned}
\tag{1}
$$

where FFN is a fully connected layer and the Multi-Head Attention can be described as:

$$
\begin{aligned}
\text{Multi-Head}(\widetilde{X}) &= [\text{head}_1 || \text{head}_2 || ... || \text{head}_h] W^O, \\
\text{head}_i &= \text{Softmax}\left( \frac{\widetilde{X}^i W^Q (\widetilde{X}^i W^K)^T}{\sqrt{d_k}} \right) \widetilde{X}^i W^V,
\end{aligned}
\tag{2}
$$

where $\widetilde{X}^i$ is the input of the $i$-th head, $W^O, W^Q, W^K$, and $W^V$ are the learnable weights matrix, $h$ is the number of the head, and $\sqrt{d_k}$ is the scaling factor.

**Spatial Contextual Attention Layer.** Inspired by Flashback [43], we introduce the Spatial Contextual Attention Layer to analyze the relationship between spatial proximity and user interactions. It assigns dynamic weights to POIs in a sequence, taking into account both the order and geographical distances, focusing on POIs most influential to future movements. The spatial weight $\omega_k$ for each POI $p_k$ in the sequence $(p_1, p_2, ..., p_k, ..., p_n)$, considering its spatial distance to all previous POIs $p_1 \sim p_k$, is:

$$
\omega_k = \sum_{j=1}^{k} \left( e^{-\beta(\Delta(p_j, p_k))} + \epsilon \right),
\tag{3}
$$

where $\Delta(p_j, p_k)$ is the haversine distance between $p_j$ and $p_k$, $\beta$ is the distance decay weight, and $\epsilon$ is a small constant to prevent division by zero. Considering $\tilde{z}_k$, an element from the Transformer output sequence $\widetilde{Z} = (\tilde{z}_1, \tilde{z}_2, ..., \tilde{z}_k, ..., \tilde{z}_n)$. The refined output $\tilde{z}_k'$ is obtained by applying spatial weight to the cumulative previous outputs, defined as follows:

$$
\tilde{z}_k' = \frac{\sum_{j=1}^{k} \omega_j \cdot \tilde{z}_j}{\sum_{j=1}^{k} \omega_j}.
\tag{4}
$$

**Prediction Layer.** To provide personalized predictive outcomes and ensure accurate representation even for users with fewer check-ins, we further introduce user embeddings $E_u^U$ and fuse it with refined output $\widetilde{Z}'$ to form the input $\mathcal{O} = [\widetilde{Z}' || E_u^U]$ for the final fully connected layer $L = \mathcal{O}W + b^P$, where $W \in \mathbb{R}^{(d_p + d_u) \times |P|}$ is the weight matrix of the fully connected layer, $b \in \mathbb{R}^{|P|}$ is the bias, and $L \in \mathbb{R}^{n \times |P|}$ is the logit scores for $n$ steps of POI prediction. As the POI prediction is essentially a sequence classification task, we adopt the standard cross-entropy loss $\mathcal{L}_{CE}$ as follows:

$$
\mathcal{L}_{CE} = -\frac{1}{N} \sum_{k=1}^{N} \sum_{i=1}^{|P|} y_i^k \log \left( \frac{\exp(l_i^k)}{\sum_{j=1}^{|P|} \exp(l_j^k)} \right),
\tag{5}
$$

where $l_i^k \in \mathbb{R}^1$ represents the logit score of the $k$-th sample for the $i$-th POI candidate in $P$, and $y_i^k$ is the ground-truth indicator on POI label $i$ for the $k$-th sample. In our implementation, we mix the $n$ steps of prediction and $B$ samples in one batch together as $N = n \times B$ samples in total.

## 4.2 Long-Tailed Graph Adjustment

In the next POI prediction task, we model user-POI interactions via a User-POI Interaction Graph $\mathcal{G}^{In}$ = $(\mathcal{V}^{In}, \mathcal{A}^{In})$, where $\mathcal{V}^{In} = [E^U||E^P] \in \mathbb{R}^{(|U|+|P|) \times d}$ is the input node feature matrix of the $\mathcal{G}^{In}$, $d$ = $d_p = d_u$, and $\mathcal{A}^{In} \in \mathbb{R}^{|U| \times |P|}$ is the adjacent matrix. It's a bipartite graph where user $U$ and POI $P$ nodes connect through edges symbolizing interaction frequencies or preferences. Graph Neural Networks (GNNs) [38] can leverage graphs to learn complex node representations, but performance hinges on graph quality. However, the $\mathcal{G}^{In}$ often has long-tailed distributions—most interactions are limited to few nodes with high visit frequency, which affects the quality of node embeddings and model efficacy. To tackle the long-tail problem in $\mathcal{G}^{In}$, we propose a denoising layer to prune and reduce sparse interactions caused by the distribution. This layer evaluates edge importance, retaining only beneficial edges for learning. Initially, an attention layer weights edges according to user-POI embedding interactions, processed by a multilayer perceptron (MLP) to obtain attention scores:

$$A_{ij} = \sigma(W^B \cdot \text{LeakyReLU}(W^A[E_i^U||E_j^P] + b^A) + b^B). \tag{6}$$

Here, $\sigma$ denotes the sigmoid function, ensuring that the attention scores $A_{ij}$ lie in the (0, 1) interval, $E_i^U$ and $E_j^P$ means the embedding of user and POI, $W$ represents the trainable weight matrix, and $b$ represents the bias. Based on the attention score $A_{ij}$, the denoising process applies a thresholding operation to filter out edges with scores below a threshold $\delta$, effectively reducing noise and focusing on high-quality interactions. This process aims to derive the denoised graph $\widetilde{\mathcal{G}}^{In} = (\mathcal{V}^{In}, \widetilde{\mathcal{A}}^{In})$ can be formalized as:

$$\widetilde{\mathcal{A}}_{ij}^{In} = \mathcal{A}_{ij}^{In} \cdot \mathbf{1}[A_{ij} \geq \delta], \tag{7}$$

where $\widetilde{\mathcal{A}}_{ij}^{In}$ denotes the refined edge and $\mathbf{1}[\cdot]$ is the indicator function. The threshold $\delta$ controls the sparsity of the graph, only edges with weights signifying a strong user-POI relationship are retained in $\widetilde{\mathcal{G}}^{In}$. It is worth noting that when all edges fall below $\delta$, the edge with the highest attention score is retained to prevent isolated nodes in the graph. The model then leverages the Graph Convolutional Network (GCN) [16] layer to learn the node embedding $E^{In}$ of the $\widetilde{\mathcal{G}}^{In}$, as follows:

$$E^{In} = \text{LeakyReLU}\left((D^{In})^{-\frac{1}{2}} \widetilde{\mathcal{A}}^{In} (D^{In})^{-\frac{1}{2}} \mathcal{V}^{In} W^{In}\right), \tag{8}$$

where $D^{In}$ is the degree matrix of the $\widetilde{\mathcal{A}}^{In}$, and $W^{In}$ is the graph convolution weight. It is noted that here we perform a slicing operation $E^{In} = E^{In}[|P| :]$ to select the node embedding representing the POI of $\widetilde{\mathcal{G}}^{In}$. Beyond merely focusing on direct interactions between users and POIs, we further extend our exploration to utilize all users' check-in data to uncover global mobility patterns among POIs. We build a user-independent directed Global Transition Graph $\mathcal{G}^{Tr} = (\mathcal{V}^{Tr}, \mathcal{A}^{Tr})$, where $\mathcal{V}^{Tr} \in \mathbb{R}^{|P| \times d_p}$ and $\mathcal{A}^{Tr} \in \mathbb{R}^{|P| \times |P|}$. Here, $\mathcal{V}^{Tr}$ is equal to $E^P$, and $\mathcal{A}^{Tr}$ stores the visit frequency between two different POIs. It is important to note that we do not perform a denoising process on the $\mathcal{G}^{Tr}$, as it accurately reflects the mobility patterns of all users, containing a wealth of global transition information. Similarly, we employ GCN refer to Equation (8) to learn the node embedding $E^{Tr}$ of the $\mathcal{G}^{Tr}$. Finally, we perform mean pooling to combine the two node embeddings $E^{In}$ and $E^{Tr}$, which yields the denoised POI embedding $\widetilde{E}^P = \frac{1}{2}(E^{In} + E^{Tr})$ that incorporate comprehensive user mobility patterns from interaction and transition graphs. To introduce denoised embedding in our model, we refine our input embedding sequence $X$ construction process as $X = \left[(\widetilde{E}_{p_1}^P, \widetilde{E}_{p_2}^P, ..., \widetilde{E}_{p_n}^P)||(E_{t_1}^T, E_{t_2}^T, ..., E_{t_n}^T)\right]$.

## 4.3 Long-Tailed Loss Adjustment

**Logit Score Adjustment.** Traditional classification models often mechanically employ the softmax function for outputting predictions, which may lead to an oversight of the potential discrepancies in the posterior distributions between training and testing data. To improve model discrimination, logit adjustment has been explored, which originates in the domain of face recognition [28, 52], It involves modifying the model's output layer (i.e., logits) to encourage the generation of more compact intra-class representations while increasing the distance between classes, thereby augmenting the model's capability to handle long-tailed data.

To address the long-tail problem in human next POI prediction tasks, we propose the Logit Score Adjustment module. It adjusts the logits by a factor that is inversely correlated with the frequency

of occurrence of each label, effectively dampening the influence of frequently occurring labels and amplifying that of rarer ones. The adjustment factor $\alpha_i$ for label $i$ with frequency $freq$ is given by:

$$\alpha_i = \tau \left[ 1 - \frac{\log(freq_i + \epsilon)}{\log(freq_{max} + \epsilon)} \right], \tag{9}$$

where $freq_{max}$ is the maximum label frequency observed in the dataset, $\tau$ is the logit adjustment weight and $\epsilon$ is a small constant to stabilize the logarithm operation. We can adjust final logits $\widetilde{l}_i \in \mathbb{R}^1$ based on the logits $l_i \in \mathbb{R}^1$ as $\widetilde{l}_i = l_i + \alpha_i$.

**Sample Weight Adjustment.** Based on the Equation (5), for the standard cross-entropy loss, we can find due to the nature of the softmax function, which normalizes the logits $l_i^k$ into probabilities, the model can become biased toward head classes. This imbalance means that the model's updates are predominantly driven by the head classes, as the loss from incorrectly classified examples in long-tailed classes contributes insignificantly to the overall loss. Even marginal improvements in the predictions for these long-tailed classes may contribute insignificantly to the overall loss. Therefore, it is necessary to reweight long-tailed samples, like with Focal Loss [17], which reduces the weights of well-classified samples to better focus on minority classes, but it does not explicitly consider the imbalance degree between classes in the long-tailed distribution. Unlike Focal Loss, we propose a novel Long-Tail Adjusted (LTA) loss to adaptively re-weight long-tailed samples. Specifically, for the final prediction layer, we have the hidden inputs of $N$ samples $\mathcal{O} = (o^1, o^2, ..., o^N)$ and the weights $W = (w^1, w^2, ..., w^{|P|})$ for $|P|$ candidates, where $o^k \in \mathbb{R}^{(d_u + d_p)}$ is from the $k$-th sample. The true class label for the $k$-th sample is denoted by $y_k$. We can take $w^{y_k} \in \mathbb{R}^{(d_u + d_p)}$ as the class "center" for the class to which the $k$-th sample truly belongs. Then we assess the impact posed by the $k$-th sample to the overall prediction through the cosine similarity between $o^k$ and $w^{y_k}$ as follows:

$$cos(o^k, w^{y_k}) = \frac{o^k \cdot w^{y_k}}{\|o^k\| \|w^{y_k}\|}. \tag{10}$$

Based on these cosine similarities, we compute the adjusted vector magnitude $\xi^k$ for each sample as:

$$\xi^k = \begin{cases} 1, & cos(o^k, w^{y_k}) > 0, \\ 1 - cos(o^k, w^{y_k}), & cos(o^k, w^{y_k}) \leq 0. \end{cases} \tag{11}$$

Then we determine the geometric mean of the vector magnitude to serve as a baseline magnitude $\bar{\xi}$. The traditional definition of the geometric mean of the vector magnitudes is the $N$-th root of their product $\bar{\xi} = \sqrt[N]{\xi^1 \xi^2 \cdots \xi^N}$. However, it can be problematic in practice due to numerical underflow or overflow when dealing with very small or very large values. To mitigate this issue, we utilize logarithm to turn the product into a sum, making the calculation more numerically stable, as follows:

$$\bar{\xi} = \exp \left( \frac{1}{N} \sum_{k=1}^{N} \log(\xi^k + \epsilon) \right). \tag{12}$$

We calculate adaptive weights $\phi^k$ for each sample using the deviation of vector magnitude from the geometric mean:

$$\phi^k = \begin{cases} 1, & \xi^k - \bar{\xi} \leq 0, \\ 1 + \xi^k - \bar{\xi}, & \xi^k - \bar{\xi} > 0. \end{cases} \tag{13}$$

:Finally, the overall Long-Tail Adjusted loss $\mathcal{L}_{LTA}$ can be formulated as:

$$\mathcal{L}_{LTA} = -\frac{1}{N} \sum_{k=1}^{N} \phi^k \sum_{i=1}^{|P|} y_i^k \log \left( \frac{\exp(\widetilde{l}_i^k)}{\sum_{j=1}^{|P|} \exp(\widetilde{l}_j^k)} \right). \tag{14}$$

By combining the Logit Score Adjustment and the Sample Weight Adjustment, we present a nuanced approach to recalibrating the model's focus across the spectrum of label frequencies. It ensures that each sample contributes to the model's learning process in proportion to its significance, as dictated by the distributional characteristics of the dataset and the discriminative capacity of the model.

## 4.4 Model Optimization

Building upon our Long-Tailed Loss Adjustment module, we further embrace auxiliary prediction tasks to optimize LoTNext. To incorporate these tasks, we define a joint loss function that combines three distinct loss components: the standard cross-entropy loss ($\mathcal{L}_{CE}$), the Long-Tail Adjusted Loss ($\mathcal{L}_{LTA}$), and the Mean Squared Error loss for auxiliary time prediction ($\mathcal{L}_{Aux}$). Each component serves a critical role: $\mathcal{L}_{CE}$ ensures the fidelity of the next POI prediction, $\mathcal{L}_{LTA}$ addresses the long-tailed data imbalance through adaptive weighting, and $\mathcal{L}_{Aux}$ measures the accuracy of the timing predictions, an auxiliary task that supports the model by providing it with temporal context, thereby improving prediction accuracy and robustness, which can be denoted as:

$$\mathcal{L}_{Aux} = \frac{1}{N} \sum_{k=1}^{N} ||\hat{t}^k - t^k||^2, \tag{15}$$

where $\hat{t}^k$ is the forecasted time slot of $k$-th candidate POI and $t^k$ is the ground truth time slot. The overall loss function is constructed as a weighted sum of these components, with the weights $\lambda$ being learnable parameters, as follows:

$$\mathcal{L}_{Joint} = \lambda_1 \mathcal{L}_{CE} + \lambda_2 \mathcal{L}_{LTA} + \lambda_3 \mathcal{L}_{Aux}. \tag{16}$$

## 5 Experiments

**Datasets & Baselines.** We evaluate our LoTNext on two publicly available real-world LBSN datasets: Gowalla and Foursquare[2] Each user check-in record includes the User ID, POI ID, latitude, longitude, and timestamp. To focus solely on the impact of long-tailed POIs and ensure the dataset's quality, we filter out inactive users with fewer than 100 check-ins. We then split each user's check-in records according to temporal order, using the first 80% for training and the remaining 20% for testing. To batch training, we uniformly segment the length of each input trajectory (e.g., 20). The specific statistical results are shown in Table 1, where we additionally calculated the percentage of POIs with a frequency smaller than 200 times and smaller than 100 times out of the total number of POIs. For instance, defining long-tailed POIs as those with a frequency of less than 100 times, approximately 98.38% of POIs could be considered long-tailed POIs. Considering both Table 1 and Table 2, the reason why the model performs about 20% points better on Foursquare compared to Gowalla is due to the more severe long-tail effect on the Gowalla dataset, along with a sparser density of the dataset.

Table 1: Basic dataset statistics.

| Dataset | Gowalla | Foursquare |
|---|---|---|
| Duration | 2009.02-2010.10 | 2012.04-2014.01 |
| #Users | 7,768 | 45,343 |
| #POIs | 106,994 | 68,879 |
| #Check-ins | 1,823,598 | 9,361,228 |
| #Trajectories | 84,357 | 429,071 |
| Density | 0.002194 | 0.002997 |
| POI frequency <200 (%) | 99.57% | 89.26% |
| POI frequency <100 (%) | 98.38% | 63.70% |

To demonstrate the performance of the LoTNext, we implement the following 10 state-of-the-art methods as the comparison baselines:

- **ST-RNN** [18] extends the RNN by introducing the spatial and temporal transition matrices.

- **DeepMove** [6] considers long-term and short-term interests of users by attention mechanism.

- **LBSN2Vec** [42] introduces the hypergraph and calculates the similarity of users and time embeddings to rank POIs.

- **LightGCN** [9] simplifies the structure of Graph Convolutional Network (GCN) to learn user preferences for POIs.

---

[2]https://sites.google.com/site/yangdingqi/home/foursquare-dataset

Table 2: Acc@k and MRR performance comparison on Gowalla and Foursquare datasets.

| Model | Gowalla | | | | Foursquare | | | |
|---|---|---|---|---|---|---|---|---|
| | Acc@1 | Acc@5 | Acc@10 | MRR | Acc@1 | Acc@5 | Acc@10 | MRR |
| ST-RNN [18] | 0.0900 | 0.2120 | 0.2730 | 0.1508 | 0.2290 | 0.4310 | 0.5050 | 0.3248 |
| DeepMove [6] | 0.0625 | 0.1304 | 0.1594 | 0.0982 | 0.2400 | 0.4319 | 0.4742 | 0.3270 |
| LBSN2Vec [42] | 0.0864 | 0.1186 | 0.1390 | 0.1032 | 0.2190 | 0.3955 | 0.4621 | 0.2781 |
| LightGCN [9] | 0.0428 | 0.1439 | 0.2115 | 0.1224 | 0.0540 | 0.1790 | 0.2710 | 0.1574 |
| LSTPM [29] | 0.0721 | 0.1843 | 0.2327 | 0.1306 | 0.2484 | 0.4489 | 0.5018 | 0.3365 |
| Flashback [43] | 0.1158 | 0.2754 | 0.3479 | 0.1925 | 0.2496 | 0.5399 | 0.6326 | 0.3805 |
| STAN [23] | 0.0891 | 0.2096 | 0.2763 | 0.1523 | 0.2265 | 0.4515 | 0.5310 | 0.3420 |
| GETNext [45] | 0.1419 | 0.3270 | 0.4081 | 0.2294 | 0.2646 | 0.5640 | 0.6431 | 0.3988 |
| Graph-Flashback [27] | 0.1495 | 0.3399 | 0.4242 | 0.2401 | 0.2786 | 0.5733 | 0.6501 | 0.4109 |
| SNPM [46] | 0.1593 | 0.3514 | 0.4346 | 0.2505 | 0.2899 | 0.5967 | 0.6763 | 0.4278 |
| **LoTNext (Ours)** | **0.1668** | **0.3605** | **0.4429** | **0.2591** | **0.3155** | **0.6059** | **0.6812** | **0.4469** |

- **LSTPM** [29] proposes geo-nonlocal LSTM to further extend DeepMove structure.

- **Flashback** [43] searches the most similar hidden states in historical information based on the current context information and updates the model.

- **STAN** [23] explores the influence between non-adjacent check-in records in trajectory sequences through the attention mechanism.

- **GETNext** [45] introduces the global mobility patterns of all users into the Transformer architecture to improve model prediction effects.

- **Graph-Flashback** [27] combines Spatial-Temporal Knowledge Graph with the sequential model to enrich the representation of each POI.

- **SNPM** [46] learns the general characteristics of POIs by constructing a POI similarity graph and aggregating similar POIs.

**Metrics.** To evaluate the model performance, we utilize two of the most common metrics for the next POI prediction: Accuracy@k (Acc@k) and Mean Reciprocal Rank (MRR). Acc@k effectively measures whether the true label is present within the top-k predicted results. Here, we consider k=1, 5, and 10 to comprehensively assess the model's performance. MRR directly quantifies the average rank of the correct label among all predictions when the correct label is not within the top-k predictions, with higher values indicating better average prediction performance by the model.

**Settings.** We implement LoTNext using PyTorch 1.13.1 on a Linux server equipped with 384GB RAM, 10-core Intel(R) Xeon(R) Silver 4210R CPU @ 2.40GHz, and Nvidia RTX 3090 GPUs. The embedding dimensions for POIs and users are set to 10, and the time embedding dimension is set to 6. For the Transformer architecture, we incorporate two multi-head attention mechanisms and 2 encoder blocks. For the spatial decay rate $\beta$, we follow the settings of Flashback [43].

**Overall Performance.** Table 2 shows the predictive performance of all baseline methods and LoTNext on two datasets. Based on Table 2, we can draw the following conclusions:

- On both public datasets, LoTNext outperforms all other state-of-the-art baseline methods across all metrics. Compared to the most recent and best-performing baseline method, SNPM, LoTNext achieves more significant improvements in Acc@1. These results indicate that LoTNext is better at predicting long-tailed POIs that are less popular but highly relevant to specific users.

- Utilizing graphs to model all user mobility patterns, thereby improving POI embeddings, representing as SNPM, Graph-Flashback, and GETNext, significantly outperform sequential methods represented by LSTPM and DeepMove, which rely solely on an individual user's short-term and long-term interests to predict the user's next location. However, the raw User-POI Interaction Graph has a large number of long-tailed nodes with a degree of 1 or very small. LoTNext, through its long-tailed graph adjustment module, effectively filters these long-tailed nodes, thereby enhancing the model's predictive performance.

**Performance on Long-Tailed Samples.** To evaluate whether our model achieves accuracy improvement on long-tailed samples, we define samples with a frequency less than 100 on Gowalla dataset

as long-tailed samples to test the model's specific predictive performance on both long-tailed and head samples. We compare LoTNext with Graph-Flashback which provides the pre-trained model for ease of comparison. As shown in Figure 3(a) and Figure 3(b), LoTNext consistently outperforms Graph-Flashback on both Acc@1 and MRR metrics, whether for head or long-tailed samples. Furthermore, Figure 3(c) reveals a notable distinction in the prediction of long-tailed POIs, LoTNext exhibits a roughly 6% higher propensity to predict long-tailed POIs compared to Graph-Flashback. This increment not only underscores the enhanced capacity of LoTNext to identify and anticipate long-tailed POIs but also demonstrates the efficacy of our methodology.

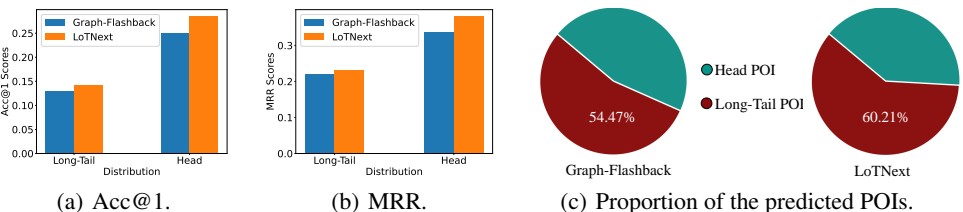

(a) Acc@1.      (b) MRR.      (c) Proportion of the predicted POIs.

Figure 3: The performance comparison of the long-tailed and head POIs between LoTNext and Graph-Flashback on Gowalla dataset.

**Ablation Study.** To analyze the impact of different modules on LoTNext, we conducted the following ablation settings: (1) without the Long-Tailed Graph Adjustment module (w/o LTGA), where we conducted with the raw graph without graph adjustment. (2) without the Long-Tailed Loss Adjustment module (w/o LTLA), meaning we only used the original cross-entropy loss for testing. (3) without the original cross-entropy loss (w/o $\mathcal{L}_{CE}$), meaning we only use Long-Tailed Loss Adjustment module ($\mathcal{L}_{LTA}$ loss). (4) without the auxiliary prediction task module, we utilized the LTLA module and cross-entropy loss function, removing the auxiliary time prediction task, denoted as w/o $\mathcal{L}_{Aux}$.

Table 3: The performance comparison among the LoTNext and variants without some components.

| Model | Gowalla | | | | Foursquare | | | |
|---|---|---|---|---|---|---|---|---|
| | Acc@1 | Acc@5 | Acc@10 | MRR | Acc@1 | Acc@5 | Acc@10 | MRR |
| w/o LTGA | 0.1617 | 0.3568 | 0.4419 | 0.2550 | 0.3020 | 0.6002 | 0.6783 | 0.4368 |
| w/o LTLA | 0.1544 | 0.3439 | 0.4266 | 0.2450 | 0.3014 | 0.5985 | 0.6758 | 0.4362 |
| w/o $\mathcal{L}_{CE}$ | 0.1550 | 0.3455 | 0.4287 | 0.2462 | 0.3029 | 0.5989 | 0.6771 | 0.4365 |
| w/o $\mathcal{L}_{Aux}$ | 0.1609 | 0.3567 | 0.4344 | 0.2523 | 0.3039 | 0.5993 | 0.6769 | 0.4370 |
| **LoTNext** | **0.1668** | **0.3605** | **0.4429** | **0.2591** | **0.3155** | **0.6059** | **0.6812** | **0.4469** |

From Table 3, we have the following findings: (1) The embeddings obtained after the LTGA module contribute to the model's predictive performance. This is mainly because long-tailed POIs can be considered noise to some extent, and appropriately eliminating some noise helps with model prediction. (2) Utilizing only the original cross-entropy loss results in performance below SNPM, indicating that the strategy of considering the long-tailed distribution through the LTLA module is effective for improving model accuracy in identifying the most relevant items. (3) The results using only the $\mathcal{L}_{LTA}$ loss show slightly higher metrics than results w/o LTLA, which suggests that the model may over-focus on long-tail data, leading to a decline in the recommendation performance for head data. For this reason, we consider incorporating the $\mathcal{L}_{CE}$ to balance the recommendation performance between long-tail data and head data. (4) Without the time prediction task, we observe a decline in the MRR metric, suggesting that temporal features play a crucial role in helping the model capture the dynamic changes in user behavior.

**Case Study: Learned POI Embedding.** Figure 4 presents t-SNE visualizations of the embeddings for the four least frequently occurring POIs on Gowalla dataset. In Figure 4(b) representing LoTNext the embeddings of these low-frequency POIs are more distinct and well-separated, indicating that LoTNext effectively captures the unique characteristics of these tail POIs. This clear separation demonstrates that LoTNext can learn meaningful representations even for the least frequent POIs, which is crucial for accurate prediction and recommendation. In contrast, Figure 4(a) showing Graph-Flashback's performance, reveals more overlapping and less distinct clusters for these low-frequency

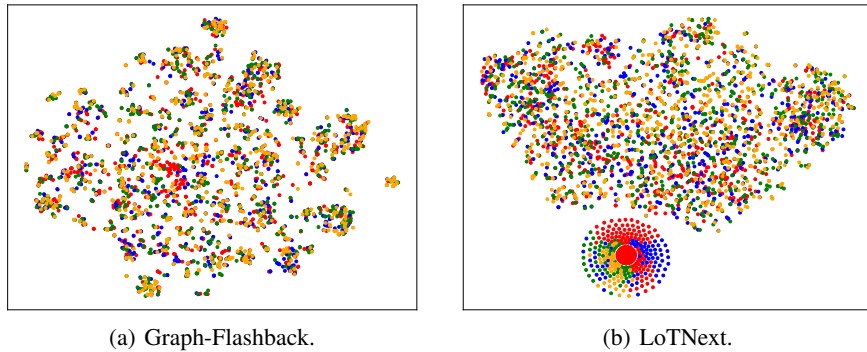

(a) Graph-Flashback.              (b) LoTNext.

Figure 4: The visualization of tail POIs on Gowalla dataset. The color represents the POI frequency.

POIs. This overlap suggests that Graph-Flashback struggles to differentiate between the tail POIs, potentially leading to less accurate predictions for these rarely visited locations.

**Case Study: Prediction on Long-Tailed Sample.**
Figure 5 provides a visual comparison of sample predictions made by the Graph-Flashback and LoTNext models on a trajectory from the Gowalla dataset for user 5. Each POI in the user's trajectory is identified by a unique ID and its visitation frequency, where the number in parentheses represents the frequency of visits. In this specific trajectory, user 5 visits a sequence of POIs. For the given POI 934, LoTNext accurately predicts the next POI to be 933, a long-tail POI with a visitation frequency of 94. In contrast, the Graph-Flashback model incorrectly predicts the next POI to be 61, a head POI with an extremely high visitation frequency of 2023. This is the same sample as the problem

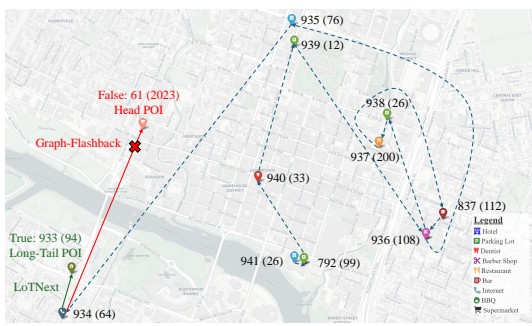

Figure 5: Sample prediction from Gowalla dataset with Graph-Flashback and LoTNext.

shown in Figure 1, demonstrating the efficacy of LoTNext in capturing the user's actual movement pattern, which encompasses both frequently and infrequently visited POIs.

## 6 Conclusion

In this work, we propose LoTNext, a novel framework for human next POI prediction under long-tailed data distribution. Specifically, we employ a Long-Tailed Graph Adjustment module to mitigate the impact of long-tailed nodes within the User-POI Interaction Graph. Additionally, to balance the influence of long-tailed data in the loss, we propose the Long-Tailed Loss Adjustment module to adjust the model's predicted logits and adaptively increase the weight of long-tailed samples. Moreover, we leverage the auxiliary prediction task to achieve spatial and temporal prediction synergy. Through comparisons with 10 state-of-the-art methods, we demonstrate the superiority of LoTNext over the most advanced approaches. A limitation of our approach lies in that LoTNext's reliance on extensive user trajectory data poses a potential risk for privacy breaches if deployed by certain institutions or companies, which could lead to negative social impacts. We plan to address it in future work.

## Acknowledgments and Disclosure of Funding

This work was supported by JST SPRING Grant Number JPMJSP2108, JSPS KAKENHI Grant Number JP24K02996, JST CREST Grant Number JPMJCR21M2 including AIP challenge program, and Initiative on Recommendation Program for Young Researchers and Woman Researchers, Information Technology Center, The University of Tokyo.

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

# A  Appendix / supplemental material

## A.1  Notations

The notations used in our paper are summarized as follows.

Table 4: Notation Table.

| Symbol | Meaning |
|---|---|
| $U, u$ | User set and user |
| $P, p$ | POI set and POI |
| $T, t$ | Time slot set and time slot |
| $E^U, E^P, E^T, E_{pos}$ | Embedding of user, POI, timestamp, and position |
| $X, \widetilde{X}$ | Embedding sequence w/o./with positional embedding |
| $Z, \widetilde{Z}$ | Output of the multi-head attention layer and transformer |
| $\widetilde{Z}', \widetilde{z}'$ | Refined output by spatial contextual attention layer |
| $\mathcal{O}, o$ | Input of the final fully connected layer |
| $L$ | Logit scores calculated by fully connected layer |
| $W, w$ | Trainable weight matrix |
| $N$ | The number of sample |
| $B$ | Batchsize |
| $\mathcal{G}^{In}$ | User-POI Interaction Graph |
| $\mathcal{G}^{Tr}$ | Global Transition Graph |
| $\mathcal{V}^{In}, \mathcal{A}^{In}$ | Input node feature matrix and adjacent matrix of $\mathcal{G}^{In}$ |
| $\mathcal{V}^{Tr}, \mathcal{A}^{Tr}$ | Input node feature matrix and adjacent matrix of $\mathcal{G}^{Tr}$ |
| $A_{ij}$ | Attention scores |
| $\widetilde{\mathcal{G}}^{In}, \widetilde{\mathcal{A}}^{In}$ | Refined User-POI Interaction Graph and its adjacent matrix |
| $E^{In}, E^{Tr}$ | Node embedding of $\mathcal{G}^{In}$ and $\mathcal{G}^{Tr}$ |
| $D^{In}, D^{Tr}$ | Degree matrix of $\mathcal{G}^{In}$ and $\mathcal{G}^{Tr}$ |
| $\widetilde{E}^P$ | Denoised POI embedding |
| $d_p, d_t, d_u$ | Hidden dimension of the POI, time and user |
| $n$ | Sequence length |
| $b$ | Bias |
| $y_i$ | Ground-truth indicator |
| $l, l'$ | Logits and adjusted logits for each POI class |
| $t_i, \hat{t}_i$ | Time slot and the forecasted time slot |
| $\omega$ | Spatial weight |
| $\beta$ | Distance decay weight |
| $\Delta(i, j)$ | Haversine distance between $p_i$ and $p_j$ |
| $\epsilon$ | Small constant |
| $\sigma$ | Sigmoid function |
| $\delta$ | Denoising threshold |
| $\tau$ | Logit adjustment weight |
| $\alpha$ | Adjustment factor |
| $\xi, \widetilde{\xi}$ | Adjusted vector magnitude and its geometric mean |
| $\phi$ | Adaptive weights for each sample |
| $\lambda$ | Learnable loss weights |

## A.2  Computational Cost

In this section, we explore the computational cost of LoTNext. We selected three sequence-based and three graph-based baselines to demonstrate the computational efficiency of our approach. Table 5 lists the inference time for each deep learning model during the testing phase (running one training/testing instance, i.e., test time divided by batch size). We ensured that all models were executed on the same RTX 3090 GPU. Surprisingly, due to batch training, the graph-based methods generally run significantly faster than the sequence-based methods. DeepMove is the fastest among the sequence-based methods, as it only considers calculating attention using historical trajectories. Compared to DeepMove, LSTPM further introduces a geographical relationship adjacency matrix to enrich the spatial context, making it slightly slower than DeepMove. STAN employs a dual-layer attention

architecture, with one attention layer aggregating spatiotemporal correlations within user trajectories and the other selecting the most likely next POI based on weighted check-ins, resulting in the longest inference time for STAN.

Table 5: Comparison of computational cost. Each method is benchmarked on the same NVIDIA GeForce RTX 3090 GPU.

| Method | Inference Time ($10^{-3}$ Seconds) |
|---|---|
| DeepMove (Sequence-based) | 1.422 |
| LSTPM (Sequence-based) | 3.417 |
| STAN (Sequence-based) | 2887.809 |
| GETNext (Graph-based) | 3.824 |
| Graph-Flashback (Graph-based) | 0.0918 |
| SNPM (Graph-based) | 0.491 |
| LoTNext (Graph-based) | 0.257 |

In the graph-based methods, GETNext introduces additional computational overhead due to the need for extra POI candidate probability reorganization based on transition attention during the final prediction stage. SNPM requires extra computation time due to the search for similar neighborhoods within the graph. As for our LoTNext, it requires more time to run compared to Graph-Flashback because LoTNext includes graph denoising and an auxiliary temporal prediction task. However, Table 2 and Table 3 demonstrate the effectiveness of our proposed modules, even at the cost of some computational time. Thus, considering that LoTNext encompasses more processing steps and overall accuracy, the increase in inference time is still acceptable.

### A.3 Hyperparameter Analysis

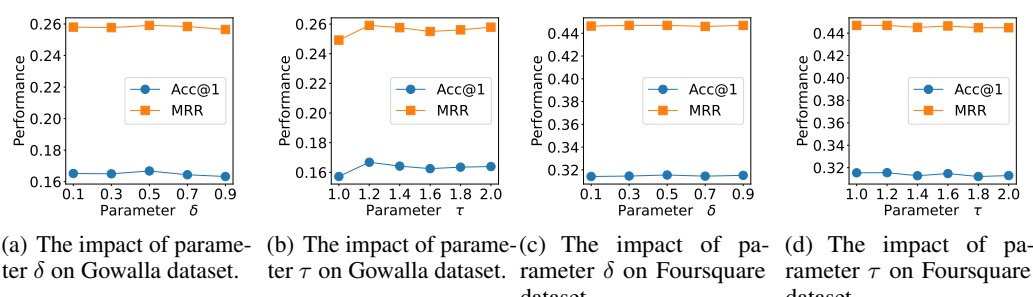

(a) The impact of parameter $\delta$ on Gowalla dataset. (b) The impact of parameter $\tau$ on Gowalla dataset. (c) The impact of parameter $\delta$ on Foursquare dataset. (d) The impact of parameter $\tau$ on Foursquare dataset.

Figure 6: Impact of denoising thresholds $\delta$ and logit adjustment weight $\tau$.

We conduct hyperparameter sensitivity experiments on the Long-Tailed Graph Adjustment module's threshold $\delta$ and the weight $\tau$ of the logit adjustment module to identify the optimal parameter values on Gowalla and Foursquare datasets. We first experiment with a range of thresholds $\delta$ from 0.1 to 0.9 in increments of 0.2, which controls the sensitivity of the model to the long-tailed distribution by filtering less significant edges in the graph. The results, shown in Figure 6(a) for Gowalla and Figure 6(c) for Foursquare, indicate that Acc@1 and MRR remain stable across different values, with the optimal threshold identified as $\delta = 0.5$. Next, we vary the logit adjustment weight $\tau$ from 1 to 2 in increments of 0.2 to test the model's performance in balancing class imbalances. Figure 6(b) and Figure 6(d) reveal that $\tau = 1.2$ yields the best results on both datasets, suggesting a moderate adjustment weight helps generalize better without overly amplifying rare classes. These consistent findings across both datasets underscore the robustness of $\delta = 0.5$ and $\tau = 1.2$, highlighting the importance of hyperparameter tuning in improving model accuracy and ranking metrics for better prediction of user behavior in diverse datasets.

## A.4 Model Training Pseudo-code

Algorithm 1 shows the pseudo-code of the LoTNext training process. In our experiments, all training instances are processed through mini-batches.

---

**Algorithm 1** Pseudo-code of training LoTNext

---

1: **Input:** User set $U$, POI set $P$, user check-in sequences $Q_u$ for each user $u \in U$
2: **Output:** Trained model parameters $\gamma$
3: $\gamma \leftarrow$ Initialize randomly
4: **while** not converge **do**
5:      Construct graph $G^{In}$ and $G^{Tr}$, apply denoising, and learn node embeddings $E^{In}$ and $E^{Tr}$ by Eq. (6)-(8)
         ▷ Graph Adjustment
6:      Compute denoised POI embedding $\tilde{E}^P$ based on $E^{In}$ and $E^{Tr}$          ▷ Embedding Denoising
7:      Calculate time and user embeddings $E^T$, $E^U$, and construct embedding sequence $X$      ▷ Embedding Initialization
8:      Transform input $X$ to $\tilde{X}$          ▷ Positional Encoding
9:      Calculate Transformer encoder output $\tilde{Z}$ by Eq. (1)          ▷ Transformer Encoder
10:      **for** each POI $p_k$ in sequence **do**
11:          Calculate spatial weight $\omega_k$ by Eq. (3)          ▷ Spatial Weight Calculation
12:          Refine output $\tilde{z}_k$ by Eq. (4)          ▷ Output Refinement
13:      **end for**
14:      Calculate fused output $O$ and logits $L$          ▷ Prediction Layer
15:      Compute cross-entropy loss $L_{CE}$ by Eq. (5)          ▷ Loss Calculation
16:      Adjust logits using $\alpha_i$ and recompute logits $\tilde{l}_i$ by Eq. (9)          ▷ Logit Adjustment
17:      Calculate adaptive weights $\phi^k$ and overall loss $L_{LTA}$ by Eq. (10)-(14)          ▷ Loss Adjustment
18:      Compute auxiliary loss $L_{Aux}$ by Eq. (15)          ▷ Auxiliary Loss
19:      Update parameters $\gamma$ by minimizing joint loss $L_{Joint}$ by Eq. (16)          ▷ Parameter Update
20: **end while**
21: **return** $\gamma$

---

# NeurIPS Paper Checklist

1. **Claims**

   Question: Do the main claims made in the abstract and introduction accurately reflect the paper's contributions and scope?

   Answer: [Yes]

   Justification: The abstract and introduction effectively summarize the paper's main contributions and the scope of the study, which focus on improving POI prediction through the proposed LoTNext method. This method addresses challenges related to the long-tailed distribution of location visitations. The main claims include the development of the Long-Tailed Graph Adjustment module and the Long-Tailed Loss Adjustment module, along with the incorporation of auxiliary prediction tasks to enhance model generalization and accuracy. These elements are precisely reflected in the detailed methodology and results sections of the paper, where the effectiveness of LoTNext is demonstrated on real-world datasets, significantly outperforming existing methods. Thus, the abstract and introduction provide a clear and accurate preview of the paper's technical content and its contributions to the field.

   Guidelines:

   - The answer NA means that the abstract and introduction do not include the claims made in the paper.
   - The abstract and/or introduction should clearly state the claims made, including the contributions made in the paper and important assumptions and limitations. A No or NA answer to this question will not be perceived well by the reviewers.
   - The claims made should match theoretical and experimental results, and reflect how much the results can be expected to generalize to other settings.
   - It is fine to include aspirational goals as motivation as long as it is clear that these goals are not attained by the paper.

2. **Limitations**

   Question: Does the paper discuss the limitations of the work performed by the authors?

   Answer: [Yes]

   Justification: The paper explicitly discusses the limitations of the proposed method, LoTNext. It acknowledges that the model does not account for emergent events such as vehicle accidents and extreme weather conditions, which are typical examples of long-tail data that can affect human mobility and, consequently, the model's robustness. Additionally, the paper highlights the privacy concerns associated with the extensive use of user trajectory data, acknowledging the potential risks if the model is deployed by certain entities. This discussion in the paper ensures that readers are fully aware of the conditions under which the model was tested and its potential limitations in real-world scenarios.

   Guidelines:

   - The answer NA means that the paper has no limitation while the answer No means that the paper has limitations, but those are not discussed in the paper.
   - The authors are encouraged to create a separate "Limitations" section in their paper.
   - The paper should point out any strong assumptions and how robust the results are to violations of these assumptions (e.g., independence assumptions, noiseless settings, model well-specification, asymptotic approximations only holding locally). The authors should reflect on how these assumptions might be violated in practice and what the implications would be.
   - The authors should reflect on the scope of the claims made, e.g., if the approach was only tested on a few datasets or with a few runs. In general, empirical results often depend on implicit assumptions, which should be articulated.
   - The authors should reflect on the factors that influence the performance of the approach. For example, a facial recognition algorithm may perform poorly when image resolution is low or images are taken in low lighting. Or a speech-to-text system might not be used reliably to provide closed captions for online lectures because it fails to handle technical jargon.

- The authors should discuss the computational efficiency of the proposed algorithms and how they scale with dataset size.
- If applicable, the authors should discuss possible limitations of their approach to address problems of privacy and fairness.
- While the authors might fear that complete honesty about limitations might be used by reviewers as grounds for rejection, a worse outcome might be that reviewers discover limitations that aren't acknowledged in the paper. The authors should use their best judgment and recognize that individual actions in favor of transparency play an important role in developing norms that preserve the integrity of the community. Reviewers will be specifically instructed to not penalize honesty concerning limitations.

3. **Theory Assumptions and Proofs**

   Question: For each theoretical result, does the paper provide the full set of assumptions and a complete (and correct) proof?

   Answer: [NA]

   Justification: The paper does not include theoretical results or formal proofs as it primarily focuses on the application of deep learning techniques to the problem of the next POI prediction. The mathematical formulations presented within the paper represent specific operational processes within the deep learning model rather than propositions requiring formal proofs. These formulations are used to describe the architecture and function of the proposed LoTNext method, including its components like the Long-Tailed Graph Adjustment module and the Long-Tailed Loss Adjustment module. Since the paper's contributions are empirical and methodological rather than theoretical, it does not involve the derivation of theorems or the necessity of providing rigorous proofs.

   Guidelines:
   - The answer NA means that the paper does not include theoretical results.
   - All the theorems, formulas, and proofs in the paper should be numbered and cross-referenced.
   - All assumptions should be clearly stated or referenced in the statement of any theorems.
   - The proofs can either appear in the main paper or the supplemental material, but if they appear in the supplemental material, the authors are encouraged to provide a short proof sketch to provide intuition.
   - Inversely, any informal proof provided in the core of the paper should be complemented by formal proofs provided in appendix or supplemental material.
   - Theorems and Lemmas that the proof relies upon should be properly referenced.

4. **Experimental Result Reproducibility**

   Question: Does the paper fully disclose all the information needed to reproduce the main experimental results of the paper to the extent that it affects the main claims and/or conclusions of the paper (regardless of whether the code and data are provided or not)?

   Answer: [Yes]

   Justification: The paper provides the detailed information required for the reproducibility of the main experimental results. It describes the datasets used, namely the Gowalla and Foursquare datasets, including the criteria for data filtering and splitting for training and testing purposes. It also specifies the baselines against which the proposed LoTNext method is compared, encompassing a comprehensive list of state-of-the-art methods. Furthermore, the paper outlines the evaluation metrics used, such as Accuracy@k and Mean Reciprocal Rank (MRR), and explains their significance in the context of the next POI prediction. Additionally, detailed descriptions of the experimental settings, including hardware specifications and software versions, are provided. This level of detail in the experimental setup, methodology, and evaluation ensures that other researchers can replicate the study and verify the claims made about the performance of the LoTNext method. The disclosure of this information supports the reproducibility of the research, adhering to the standards required for scientific verification and validation.

   Guidelines:
   - The answer NA means that the paper does not include experiments.

- If the paper includes experiments, a No answer to this question will not be perceived well by the reviewers: Making the paper reproducible is important, regardless of whether the code and data are provided or not.
- If the contribution is a dataset and/or model, the authors should describe the steps taken to make their results reproducible or verifiable.
- Depending on the contribution, reproducibility can be accomplished in various ways. For example, if the contribution is a novel architecture, describing the architecture fully might suffice, or if the contribution is a specific model and empirical evaluation, it may be necessary to either make it possible for others to replicate the model with the same dataset, or provide access to the model. In general. releasing code and data is often one good way to accomplish this, but reproducibility can also be provided via detailed instructions for how to replicate the results, access to a hosted model (e.g., in the case of a large language model), releasing of a model checkpoint, or other means that are appropriate to the research performed.
- While NeurIPS does not require releasing code, the conference does require all submissions to provide some reasonable avenue for reproducibility, which may depend on the nature of the contribution. For example
  (a) If the contribution is primarily a new algorithm, the paper should make it clear how to reproduce that algorithm.
  (b) If the contribution is primarily a new model architecture, the paper should describe the architecture clearly and fully.
  (c) If the contribution is a new model (e.g., a large language model), then there should either be a way to access this model for reproducing the results or a way to reproduce the model (e.g., with an open-source dataset or instructions for how to construct the dataset).
  (d) We recognize that reproducibility may be tricky in some cases, in which case authors are welcome to describe the particular way they provide for reproducibility. In the case of closed-source models, it may be that access to the model is limited in some way (e.g., to registered users), but it should be possible for other researchers to have some path to reproducing or verifying the results.

5. **Open access to data and code**

Question: Does the paper provide open access to the data and code, with sufficient instructions to faithfully reproduce the main experimental results, as described in supplemental material?

Answer: [Yes]

Justification: The paper grants open access to both the code and data necessary for reproducing the main experimental results, as stated in the abstract and introduction sections. The code for the proposed LoTNext method is available at a specified URL, hosted on a public repository which maintains anonymity as per submission guidelines. This repository includes detailed instructions on setting up the environment, executing the code, and reproducing the experiments conducted with the LoTNext method. Additionally, the datasets used, Gowalla and Foursquare, are publicly available and well-documented, allowing researchers to access and use them for replication purposes. Instructions for data preprocessing and setup are clearly provided, ensuring that other researchers can faithfully replicate the study's findings. This comprehensive provision of resources supports the transparency and reproducibility of the research, aligning with the conference's guidelines for open access to data and code.

Guidelines:

- The answer NA means that paper does not include experiments requiring code.
- Please see the NeurIPS code and data submission guidelines (`https://nips.cc/public/guides/CodeSubmissionPolicy`) for more details.
- While we encourage the release of code and data, we understand that this might not be possible, so "No" is an acceptable answer. Papers cannot be rejected simply for not including code, unless this is central to the contribution (e.g., for a new open-source benchmark).

- The instructions should contain the exact command and environment needed to run to reproduce the results. See the NeurIPS code and data submission guidelines (`https://nips.cc/public/guides/CodeSubmissionPolicy`) for more details.
- The authors should provide instructions on data access and preparation, including how to access the raw data, preprocessed data, intermediate data, and generated data, etc.
- The authors should provide scripts to reproduce all experimental results for the new proposed method and baselines. If only a subset of experiments are reproducible, they should state which ones are omitted from the script and why.
- At submission time, to preserve anonymity, the authors should release anonymized versions (if applicable).
- Providing as much information as possible in supplemental material (appended to the paper) is recommended, but including URLs to data and code is permitted.

6. **Experimental Setting/Details**

Question: Does the paper specify all the training and test details (e.g., data splits, hyperparameters, how they were chosen, type of optimizer, etc.) necessary to understand the results?

Answer: [Yes]

Justification: The paper provides comprehensive details about the experimental setup necessary to understand and reproduce the results. It specifies the data splits used for training and testing, with 80% of the data used for training and the remaining 20% for testing. The paper also describes the hyperparameters employed, including the dimensions for user and POI embeddings, and the configuration of the Transformer architecture used in the model.

Guidelines:

- The answer NA means that the paper does not include experiments.
- The experimental setting should be presented in the core of the paper to a level of detail that is necessary to appreciate the results and make sense of them.
- The full details can be provided either with the code, in appendix, or as supplemental material.

7. **Experiment Statistical Significance**

Question: Does the paper report error bars suitably and correctly defined or other appropriate information about the statistical significance of the experiments?

Answer: [No]

Justification: To ensure the reproducibility of our experimental results, we fixed the random seed during the data splitting and model execution phases. As a result, we did not report statistical significance measures such as error bars, confidence intervals, or statistical significance tests.

Guidelines:

- The answer NA means that the paper does not include experiments.
- The authors should answer "Yes" if the results are accompanied by error bars, confidence intervals, or statistical significance tests, at least for the experiments that support the main claims of the paper.
- The factors of variability that the error bars are capturing should be clearly stated (for example, train/test split, initialization, random drawing of some parameter, or overall run with given experimental conditions).
- The method for calculating the error bars should be explained (closed form formula, call to a library function, bootstrap, etc.)
- The assumptions made should be given (e.g., Normally distributed errors).
- It should be clear whether the error bar is the standard deviation or the standard error of the mean.
- It is OK to report 1-sigma error bars, but one should state it. The authors should preferably report a 2-sigma error bar than state that they have a 96% CI, if the hypothesis of Normality of errors is not verified.

- For asymmetric distributions, the authors should be careful not to show in tables or figures symmetric error bars that would yield results that are out of range (e.g. negative error rates).
- If error bars are reported in tables or plots, The authors should explain in the text how they were calculated and reference the corresponding figures or tables in the text.

8. **Experiments Compute Resources**

Question: For each experiment, does the paper provide sufficient information on the computer resources (type of compute workers, memory, time of execution) needed to reproduce the experiments?

Answer: [Yes]

Justification: The paper provides detailed information in Section 5 about the compute resources used for the experiments, including the type of compute workers (Linux server with Intel Xeon Silver 4210R CPU and Nvidia RTX 3090 GPUs), memory (384GB RAM), and the software environment (PyTorch 1.13.1). The specifications for the embedding dimensions, the Transformer architecture, and the spatial decay rate are also clearly described, ensuring that the experiments can be accurately reproduced.

Guidelines:

- The answer NA means that the paper does not include experiments.
- The paper should indicate the type of compute workers CPU or GPU, internal cluster, or cloud provider, including relevant memory and storage.
- The paper should provide the amount of compute required for each of the individual experimental runs as well as estimate the total compute.
- The paper should disclose whether the full research project required more compute than the experiments reported in the paper (e.g., preliminary or failed experiments that didn't make it into the paper).

9. **Code Of Ethics**

Question: Does the research conducted in the paper conform, in every respect, with the NeurIPS Code of Ethics https://neurips.cc/public/EthicsGuidelines?

Answer: [Yes]

Justification: We read the NeurIPS Code of Ethics carefully and ensure we follow all the requirements.

Guidelines:

- The answer NA means that the authors have not reviewed the NeurIPS Code of Ethics.
- If the authors answer No, they should explain the special circumstances that require a deviation from the Code of Ethics.
- The authors should make sure to preserve anonymity (e.g., if there is a special consideration due to laws or regulations in their jurisdiction).

10. **Broader Impacts**

Question: Does the paper discuss both potential positive societal impacts and negative societal impacts of the work performed?

Answer: [Yes]

Justification: The paper addresses both the potential positive and negative societal impacts of the proposed LoTNext method. On the positive side, the method aims to enhance personalized navigation, optimize recommendation systems, and facilitate urban mobility and planning. These improvements could significantly enhance the quality of life for city residents by making navigation and planning more efficient and tailored to individual needs. On the negative side, the paper acknowledges specific risks associated with the deployment of LoTNext, particularly concerning privacy. The reliance on extensive user trajectory data could potentially lead to privacy breaches if the model is employed by institutions or companies without stringent data protection measures. This could have adverse social impacts, such as unauthorized surveillance or misuse of personal data. The paper's discussion of these potential harms demonstrates an awareness of the broader implications of deploying

such technology in real-world settings. By addressing both the potential benefits and risks, the paper provides a balanced view of its societal impact, complying with ethical standards for transparency and responsibility in AI research. This acknowledgment of both positive and negative impacts ensures that readers and potential users are well-informed about the capabilities and limitations of the proposed method.

Guidelines:

- The answer NA means that there is no societal impact of the work performed.
- If the authors answer NA or No, they should explain why their work has no societal impact or why the paper does not address societal impact.
- Examples of negative societal impacts include potential malicious or unintended uses (e.g., disinformation, generating fake profiles, surveillance), fairness considerations (e.g., deployment of technologies that could make decisions that unfairly impact specific groups), privacy considerations, and security considerations.
- The conference expects that many papers will be foundational research and not tied to particular applications, let alone deployments. However, if there is a direct path to any negative applications, the authors should point it out. For example, it is legitimate to point out that an improvement in the quality of generative models could be used to generate deepfakes for disinformation. On the other hand, it is not needed to point out that a generic algorithm for optimizing neural networks could enable people to train models that generate Deepfakes faster.
- The authors should consider possible harms that could arise when the technology is being used as intended and functioning correctly, harms that could arise when the technology is being used as intended but gives incorrect results, and harms following from (intentional or unintentional) misuse of the technology.
- If there are negative societal impacts, the authors could also discuss possible mitigation strategies (e.g., gated release of models, providing defenses in addition to attacks, mechanisms for monitoring misuse, mechanisms to monitor how a system learns from feedback over time, improving the efficiency and accessibility of ML).

11. **Safeguards**

Question: Does the paper describe safeguards that have been put in place for responsible release of data or models that have a high risk for misuse (e.g., pretrained language models, image generators, or scraped datasets)?

Answer: [NA]

Justification: Our paper poses no such risks. We did not use any pre-trained models, and all datasets are sourced from publicly available data, which are commonly used in related work.

Guidelines:

- The answer NA means that the paper poses no such risks.
- Released models that have a high risk for misuse or dual-use should be released with necessary safeguards to allow for controlled use of the model, for example by requiring that users adhere to usage guidelines or restrictions to access the model or implementing safety filters.
- Datasets that have been scraped from the Internet could pose safety risks. The authors should describe how they avoided releasing unsafe images.
- We recognize that providing effective safeguards is challenging, and many papers do not require this, but we encourage authors to take this into account and make a best faith effort.

12. **Licenses for existing assets**

Question: Are the creators or original owners of assets (e.g., code, data, models), used in the paper, properly credited and are the license and terms of use explicitly mentioned and properly respected?

Answer: [Yes]

Justification: We have properly credited the creators and original owners of all assets used in the paper, including code, data, and models. The sources and relevant citations are clearly indicated in the paper. Additionally, the licenses and terms of use for these assets have

been explicitly mentioned and properly respected. We have used publicly available datasets, citing the original papers that produced them, and included the specific versions and URLs where applicable. The licenses, such as CC-BY 4.0, have been noted, ensuring compliance with the terms of use.

Guidelines:

- The answer NA means that the paper does not use existing assets.
- The authors should cite the original paper that produced the code package or dataset.
- The authors should state which version of the asset is used and, if possible, include a URL.
- The name of the license (e.g., CC-BY 4.0) should be included for each asset.
- For scraped data from a particular source (e.g., website), the copyright and terms of service of that source should be provided.
- If assets are released, the license, copyright information, and terms of use in the package should be provided. For popular datasets, `paperswithcode.com/datasets` has curated licenses for some datasets. Their licensing guide can help determine the license of a dataset.
- For existing datasets that are re-packaged, both the original license and the license of the derived asset (if it has changed) should be provided.
- If this information is not available online, the authors are encouraged to reach out to the asset's creators.

13. **New Assets**

    Question: Are new assets introduced in the paper well documented and is the documentation provided alongside the assets?

    Answer: [Yes]

    Justification: The new assets introduced in the paper are thoroughly documented. Detailed documentation is provided alongside the assets, including information about the dataset/code/model, training procedures, licenses, and limitations. This ensures that other researchers can effectively utilize and build upon our work.

    Guidelines:

    - The answer NA means that the paper does not release new assets.
    - Researchers should communicate the details of the dataset/code/model as part of their submissions via structured templates. This includes details about training, license, limitations, etc.
    - The paper should discuss whether and how consent was obtained from people whose asset is used.
    - At submission time, remember to anonymize your assets (if applicable). You can either create an anonymized URL or include an anonymized zip file.

14. **Crowdsourcing and Research with Human Subjects**

    Question: For crowdsourcing experiments and research with human subjects, does the paper include the full text of instructions given to participants and screenshots, if applicable, as well as details about compensation (if any)?

    Answer: [NA]

    Justification: Although our research involves predicting human next points of interest (POI), our data is sourced from publicly available datasets, which are commonly used in related work. The sources and relevant citations are clearly indicated in the paper, making this question not applicable to our study.

    Guidelines:

    - The answer NA means that the paper does not involve crowdsourcing nor research with human subjects.
    - Including this information in the supplemental material is fine, but if the main contribution of the paper involves human subjects, then as much detail as possible should be included in the main paper.

- According to the NeurIPS Code of Ethics, workers involved in data collection, curation, or other labor should be paid at least the minimum wage in the country of the data collector.

15. **Institutional Review Board (IRB) Approvals or Equivalent for Research with Human Subjects**

    Question: Does the paper describe potential risks incurred by study participants, whether such risks were disclosed to the subjects, and whether Institutional Review Board (IRB) approvals (or an equivalent approval/review based on the requirements of your country or institution) were obtained?

    Answer: [NA]

    Justification: Although our research involves predicting human next points of interest (POI), our data is sourced from publicly available datasets, which are commonly used in related work. The sources and relevant citations are clearly indicated in the paper, making this question not applicable to our study.

    Guidelines:
    - The answer NA means that the paper does not involve crowdsourcing nor research with human subjects.
    - Depending on the country in which research is conducted, IRB approval (or equivalent) may be required for any human subjects research. If you obtained IRB approval, you should clearly state this in the paper.
    - We recognize that the procedures for this may vary significantly between institutions and locations, and we expect authors to adhere to the NeurIPS Code of Ethics and the guidelines for their institution.
    - For initial submissions, do not include any information that would break anonymity (if applicable), such as the institution conducting the review.

