# OpenReview forum: "Taming the Long Tail in Human Mobility Prediction"
_NeurIPS.cc/2024/Conference — NeurIPS 2024 poster_

### Official Review · Reviewer_JhGF · 2024-07-12

**Soundness:** 3
**Presentation:** 3
**Contribution:** 3
**Rating:** 7
**Confidence:** 5

**Summary:**

This paper addresses the challenge of predicting less frequently visited points-of-interest (POIs) in human mobility data, a problem known as the long-tail issue in spatial distribution. The authors introduce a new framework called Long-Tailed Adjusted POI Prediction (LoTNext), which includes two main components: long-tailed graph adjustment module and long-tailed loss adjustment module. Additionally, the framework employs an auxiliary prediction task to enhance the model's generalization and overall accuracy. The effectiveness of LoTNext is demonstrated through experiments on two real-world trajectory datasets, where it significantly outperforms existing state-of-the-art methods in human mobility prediction.

**Strengths:**

1. The code has been provided, which makes the reproducibility of this paper good.
2. The paper is generally well-writern and easy to follow.
3. The proposed method is motivation-grounded.

**Weaknesses:**

1. The presentation quality of this paper can be further enhanced.
2. The authors are encouraged to conduct experiments on more datasets and provide more detailed analysis.
3.  This paper can supplement more theoretical analysis to guarantee the proposed method's effectiveness.

**Questions:**

1. I am curious if the structure of preliminary model structure can bring significant influence to the final results.
2. According to my experiments, the detailed value setting of $\lambda_1$,  $\lambda_2$, and  $\lambda_3$ in Eq. 16 can lead to the obvious variation of the ultimate model performance. However, the hyperparameter sensitivity experiments on this point is missing. More discussions and empirical results regarding this are welcomed.

**Limitations:**

See above weaknesses and questions.

---

> ### Author Rebuttal · Authors · 2024-08-06
>
> # Response to Reviewer 7hGF:
> **Weaknesses:**
> > Q1. The presentation quality of this paper can be further enhanced.
>
> **A1:** Thank you for your suggestions. We will check and polish the entire paper to ensure the presentation can be more clearly.
>
> > Q2. The authors are encouraged to conduct experiments on more datasets and provide more detailed analysis.
>
> **A2:** We appreciate your comments. We plan to introduce more datasets in our future work to more comprehensively evaluate our model. In addition, to better evaluate our model's performance, we add two latest baselines on the existing dataset. Please refer to the **"global" response** (please see above) for detailed results.
>
> > Q3. This paper can supplement more theoretical analysis to guarantee the proposed method's effectiveness.
>
> **A3:** Thank you for bringing this problem to our attention. We want to discuss **the motivation behind our sample weight adjustment module design** from a theoretical perspective.
> * For the metric Acc@1 we have: $\text{Acc} = \frac{1}{N} \sum_{i=1}^{N} \mathbf{1}(\hat{y}^i = y^i)$, where $\mathbf{1}(\cdot)$ is the indicator function. On long-tail datasets, using the arithmetic mean would result in **very small penalties for tail data with low accuracy**. For example, the average of 0.01 and 0.99 is 0.5.
> * In contrast, the harmonic mean: $H = \frac{n}{\sum_{i=1}^{n} \frac{1}{x_i}}$, which yields a value of 0.02 for 0.01 and 0.99, demonstrating **higher sensitivity to small values**, which is beneficial for penalizing tail data. However, the harmonic mean is defined by **reciprocals**, which increases the **difficulty of model optimization**.
> * Similarly, the geometric mean of 0.01 and 0.99 is 0.10, which is also very **sensitive to small values**. Its simple definition allows the model to** optimize effectively** based on it. Therefore, we finally choose geometric mean to calculate baseline magnitude.
>
> We hope these discussions can address your concerns.
>
> **Questions:**
> > Q4. I am curious if the structure of preliminary model structure can bring significant influence to the final results.
>
> **A4:** Thanks for pointing out this potential problem. We try to remove the **Spatial Contextual Attention module (SC Att.)** and replace the **Transformer** with the most traditional **RNN** model to evaluate its impact on the overall model. The results are shown below.
>
> Table Ⅰ. The different preliminary model performance comparison on Gowalla dataset.
> |Method/Metric|Acc@1|Acc@5|Acc@10|MRR|NDCG@5|NDCG@10|
> |-|-|-|-|-|-|-|
> |With RNN|0.1529|0.3400|0.4210|0.2423|0.2511|0.2773|
> |Wtihout SC Att.|0.1596|0.3547|0.4376|0.2523|0.2619|0.2888|
> |**LoTNext**|**0.1668**|**0.3605**|**0.4429**|**0.2591**|**0.2686**|**0.2953**|
>
> Table Ⅱ. The different preliminary model performance comparison on Foursquare dataset.
> |Method/Metric|Acc@1|Acc@5|Acc@10|MRR|NDCG@5|NDCG@10|
> |-|-|-|-|-|-|-|
> |With RNN|0.2532|0.5242|0.6044|0.3778|0.3997|0.4258|
> |Wtihout SC Att.|0.3117|0.6040|0.6805|0.4410|0.4740|0.4994|
> |**LoTNext**|**0.3155**|**0.6059**|**0.6812**|**0.4469**|**0.4753**|**0.5001**|
>
> * We can observe that replacing the Transformer module with an RNN significantly impacts the performance, especially on the Foursquare dataset. This highlights the powerful modeling capabilities of the Transformer.
> * The RNN model under the optimization by our graph adjustment module, loss adjustment module, and other components has resulted in superior performance compared to other RNN-based models, such as STAN and Flashback. This demonstrates that our approach is both **generalizable and effective**.
> * Removing the Spatial Contextual Attention layer has a substantial effect on the Acc@1 results, which is the most critical metric for accurately assessing the model's predictive ability.
> * Therefore, **both modules are essential for the current model**.
>
> > Q5. According to my experiments, the detailed value setting of $\lambda_1$, $\lambda_2$, $\lambda_3$ and in Eq. 16 can lead to the obvious variation of the ultimate model performance. However, the hyperparameter sensitivity experiments on this point is missing. More discussions and empirical results regarding this are welcomed.
>
> **A5:** Thanks for your inspiring questions. We **fixed the weights** of three different $\lambda$ to test the model performance, and the results are as follows.
>
> Table Ⅲ. The model performance comparison of the parameter $\lambda$ on Gowalla dataset.
> |Weight/Metric|Acc@1|Acc@5|Acc@10|MRR|NDCG@5|NDCG@10|
> |-|-|-|-|-|-|-|
> |[0.5, 0.25, 0.25]|0.1641|0.3595|0.4408|0.2580|0.2676|0.2950|
> |[0.25, 0.5, 0.25]|0.1668|0.3596|0.4424|0.2588|0.2680|0.2951|
> |[0.25, 0.25, 0.5]|0.1653|0.3594|0.4421|0.2584|0.2680|0.2952|
> |[0.33, 0.33, 0.33]|0.1580|0.3551|0.4386|0.2516|0.2614|0.2885|
> |**Dynamic**|**0.1668**|**0.3605**|**0.4429**|**0.2591**|**0.2686**|**0.2953**|
>
> Table Ⅳ. The model performance comparison of the parameter $\lambda$ on Foursquare dataset.
> |Weight/Metric|Acc@1|Acc@5|Acc@10|MRR|NDCG@5|NDCG@10|
> |-|-|-|-|-|-|-|
> |[0.5, 0.25, 0.25]|0.3082|0.6037|0.6812|0.4419|0.4689|0.4942|
> |[0.25, 0.5, 0.25]|0.3153|0.6059|0.6811|0.4469|0.4734|0.4983|
> |[0.25, 0.25, 0.5]|0.3121|0.6055|0.6810|0.4452|0.4722|0.4974|
> |[0.33, 0.33, 0.33]|0.3131|0.6053|0.6812|0.4453|0.4720|0.4970|
> |**Dynamic**|**0.3155**|**0.6059**|**0.6812**|**0.4469**|**0.4753**|**0.5001**|
>
> * The value of $\lambda$ does not significantly impact the performance.
> * When $\lambda_2$ is maximum, indicating an increased weight for our LTA loss, the performance is optimal, second only to the dynamic parameters.
> * When the three $\lambda$ values are equal, the model's performance on the Gowalla dataset shows a significant decline. This indicates that the model requires different emphasis on different losses to achieve further improvement.
>
> *If you believe our paper is qualified for NeurIPS, we would truly appreciate it if you could give us further support by increasing the score.*

---

> > ### Comment · Reviewer_JhGF · 2024-08-09
> > **Response to Author's Rebuttal**
> >
> > Thanks for your detailed response. Most of my concerns have been addressed. I will raise the score.

---

> > > ### Author Response · Authors · 2024-08-09
> > >
> > > Thank you for taking the time to respond to our rebuttal! We are happy to hear that your concerns have been solved and that our paper warrants acceptance!

---

### Official Review · Reviewer_mZab · 2024-07-12

**Soundness:** 3
**Presentation:** 3
**Contribution:** 3
**Rating:** 6
**Confidence:** 3

**Summary:**

The paper presents the Long-Tail Adjusted Next POI Prediction (LoTNext) framework to address the long-tail problem in next POI prediction. This problem refers to the uneven spatial and temporal distribution of POI visits, making it challenging for prediction models to predict less frequently visited POIs. LoTNext combines a Long-Tailed Graph Adjustment module to reduce the noise and impact of long-tailed nodes in the user-POI interaction graph and a Long-Tailed Loss Adjustment module to balance the loss between head and tail POIs. Additionally, an auxiliary prediction task is employed to enhance generalization and accuracy. The proposed method was evaluated on two real-world trajectory datasets, Gowalla and Foursquare, where it significantly outperformed existing methods.

**Strengths:**

- LoTNext introduces a unique combination of graph adjustment and loss adjustment modules to tackle the long-tail problem, which is a significant contribution to the field of human mobility prediction.
- The framework is evaluated on two real-world datasets and compared with ten existing methods, demonstrating superior performance across multiple metrics.
- The paper provides a thorough explanation of the methodology, including the embedding generation, transformer encoder, spatial contextual attention layer, and the overall optimization process, making it reproducible and transparent.

**Weaknesses:**

- The proposed model is complex and involves multiple components and adjustments, but it is not clear how computationally expensive it would be to make predictions in services and elsewhere.
- The model performed well on the dataset used, but it is unclear under what conditions the proposed method will perform well, such as visit intervals and frequency of visits.

**Questions:**

How does the complexity of LoTNext affect its scalability and real-time performance in practical applications? Are there any simplifications or optimizations that can be applied without significantly compromising performance?

**Limitations:**

The authors have adequately addressed the limitation of their work, particularly the potential privacy risks associated with the extensive use of user trajectory data.

---

> ### Author Rebuttal · Authors · 2024-08-06
>
> # Response to Reviewer mZab:
> **Weaknesses:**
> > Q1. The proposed model is complex and involves multiple components and adjustments, but it is not clear how computationally expensive it would be to make predictions in services and elsewhere.
>
> **A1:** Thank you for your insightful questions. Table 3 in the appendix lists the inference time for each deep learning model during the testing phase (running one training/testing instance, i.e., test time divided by batch size). For a more intuitive comparison of computational cost, we copied Table 3 as Table Ⅰ below. We selected three sequence-based and three graph-based baselines to demonstrate the computational efficiency of our approach. We ensured that all models were executed on the same RTX 3090 GPU.
>
> 1. Sequence-based baseline
>
> * **DeepMove** is the fastest among the sequence-based methods, as it only considers calculating attention using historical trajectories.
> * Compared to **DeepMove**, **LSTPM** further introduces a geographical relationship adjacency matrix to enrich the spatial context, making it slightly slower than **DeepMove**.
> * **STAN** employs a dual-layer attention architecture, with one attention layer aggregating spatiotemporal correlations within user trajectories and the other selecting the most likely next POI based on weighted check-ins, resulting in the longest inference time for **STAN**.
>
> 2. Graph-based baseline
> * Due to batch training, the graph-based methods generally run significantly faster than the sequence-based methods.
> * **GETNext** introduces additional computational overhead due to the need for extra POI candidate probability reorganization based on transition attention during the final prediction stage.
> * **SNPM** requires extra computation time due to the search for similar neighborhoods within the graph.
> * **LoTNext** requires more time to run compared to **Graph-Flashback** because **LoTNext** includes graph denoising and an auxiliary temporal prediction task. Table 1 and 2 (from the original paper) demonstrate the effectiveness of our proposed modules, even at the cost of some computational time. Therefore, considering that **LoTNext** encompasses more processing steps and overall accuracy, the increase in inference time is still acceptable.
>
> Table Ⅰ Comparison of computational cost. Each method is benchmarked on the same NVIDIA
> GeForce RTX 3090 GPU.
> |Method| Inference Time ($10^{-3}$ Seconds) |
> |-|-|
> |DeepMove (Sequence-based)|1.422|
> |LSTPM (Sequence-based)|3.417|
> |STAN (Sequence-based)|2887.809|
> |GETNext (Graph-based)|3.824|
> |Graph-Flashback (Graph-based)|0.0918|
> |SNPM (Graph-based)|0.491|
> |LoTNext (Graph-based)|0.257|
>
>
> > Q2. The model performed well on the dataset used, but it is unclear under what conditions the proposed method will perform well, such as visit intervals and frequency of visits.
>
> **A2:** Thanks for proposing this potential concern. We further analyzed the visit intervals for all users in the two datasets. We firstly categorized the intervals into eight hourly ranges: [0, 1], (1, 3], (3, 6], (6, 12], (12, 24], (24, 48], (48, 72], and (72, ∞). The detailed results are shown in Figure III attached in the **"global" response** PDF file (please see above).
>
> * Based on Figure III, approximately 30% of the check-in intervals in the Gowalla dataset are less than or equal to one hour, with half of the intervals being within 6 hours.
> * In contrast, the Foursquare dataset has a more evenly distributed interval range, with half of the intervals within 12 hours, and around 20% of the intervals exceeding 72 hours.
> * These **complex and uneven time distributions** highlight the importance of our time auxiliary prediction module.
>
> Regarding the frequency of visits, Figures 7(a) and 7(b) in the appendix show the relevant results.
> * We can observe that the maximum occurrence frequency of POIs in the Foursquare dataset is ten times that of the Gowalla dataset, and the number of extreme values is also higher in the Foursquare dataset.
> * Additionally, it is clear that the majority of POIs in both datasets have very low occurrence frequencies.
> * These results intuitively reflect **the long-tail nature of both datasets**.
>
> **Questions:**
> > Q3. How does the complexity of LoTNext affect its scalability and real-time performance in practical applications? Are there any simplifications or optimizations that can be applied without significantly compromising performance?
>
> **A3:** Thank you for the inspiring question. We acknowledge that using multiple deep learning methods can indeed affect the scalability and real-time performance of the model during actual deployment. However, our LoTNext model has adaptable simplification measures.
> * We can conduct **LightGCN** instead of traditional GCN can improve computation speed. The performance of **LightGCN** has been validated in many related works, ensuring it does not compromise the performance of GCN as a graph learning module.
> * We can conduct **graph pre-training** to enhance training efficiency. A **well-pretrained graph representation** might eliminate the need for our graph adjustment module without affecting the overall model performance, thereby speeding up the training process.
> * In addition, the loss adjustment module **do not involve overly complex calculations**, thus not significantly increasing computational overhead.
>
> *If the above discussions address your concerns, we would greatly appreciate it if you could consider increasing the score.*

---

> > ### Author Response · Authors · 2024-08-11
> >
> > I appreciate the feedback provided so far. Could you please share any additional thoughts or clarifications on the rebuttal? Your detailed review is important for the progress of this paper.

---

> > > ### Comment · Reviewer_mZab · 2024-08-14
> > >
> > > Thank you for your careful responses. Some of my concerns have been addressed by your explanations, and as a result, I will raise my rating to 6.

---

> > > > ### Author Response · Authors · 2024-08-14
> > > >
> > > > Thank you for your thorough review and for taking the time to reconsider your evaluation. I greatly appreciate your updated rating and your thoughtful feedback throughout this process.

---

> ### Comment · Area_Chair_odhj · 2024-08-13
>
> Dear Reviewer
>
> Could you please read the rebuttal and engage the discussion with the authors ASAP?
>
> AC

---

### Official Review · Reviewer_Et5T · 2024-07-16

**Soundness:** 3
**Presentation:** 3
**Contribution:** 3
**Rating:** 6
**Confidence:** 5

**Summary:**

This paper introduces the LoTNext framework, which is designed to improve the prediction of human mobility patterns, specifically addressing the challenge of long-tail distribution in POI visitations. The authors propose a novel approach that includes a Long-Tailed Graph Adjustment module and a Long-Tailed Loss Adjustment module, along with an auxiliary prediction task, to enhance the model's ability to predict less frequently visited POIs. The paper demonstrates the effectiveness of LoTNext through comprehensive experiments on two real-world datasets, showing significant improvements over existing state-of-the-art methods.

**Strengths:**

I like the research gap proposed by this paper. This is a worthwhile issue to study.

**Weaknesses:**

(1) The evaluation could be expanded to include a broader range of metrics to further validate the generalizability of the LoTNext framework.
(2) It's better to have more explainability related experiments.
(3) A more detailed literature review is needed (at least in the appendix) so that the novelty of the method could be better evaluated.
(4) The comparison methods used are somewhat outdated. Why didn't you use the latest methods, such as TPG (https://arxiv.org/abs/2304.04151) or LLM-Move (https://arxiv.org/pdf/2404.01855), for comparison?

**Questions:**

(1) Why does cutting off the long tail in the user-POI graph for noise reduction, followed by using loss to reintroduce the long tail, theoretically improve long tail prediction?

(2) In Figure 4, how should we interpret "four least frequently occurring pois"? Does it refer to only the four POIs with the lowest frequencies? I guess many POIs only appear once.

---

> ### Author Rebuttal · Authors · 2024-08-06
>
> # Response to Reviewer Et5T:
> **Weaknesses:**
> > Q1. The evaluation could be expanded to include a broader range of metrics to further validate the generalizability of the LoTNext framework.
>
> **A1:** Thank you for valuable suggestions. We add Normalized Discounted Cumulative Gain (NDCG) as a new metric, the specific results are shown in **"global" response** (please see above).
>
> > Q2. It's better to have more explainability related experiments.
>
> **A2:** Thanks for your comments. To make our model prediction results more interpretable, we visualized the prediction results of LoTNext and Graph-Flashback on the Gowalla dataset, as shown in Figure II attached in the **"global" response** PDF file (please see above). In Figure II, the blue circles represent the user's historical trajectories, the green circles indicate the model's predicted top-20 POIs, and the red circle marks the actual next POI.
> * From Figure II, we can observe the predicted POI are **more dispersed and cover a wider area** in the LoTNext compared to the Graph-Flashback. This indicates that LoTNext provides a broader prediction range within the user's historical trajectory, potentially capturing a wider variety of POIs.
> * The LoTNext model's predictions **include the actual next POI** within its top-20 suggestions, demonstrating its higher accuracy. In contrast, the Graph-Flashback model's predictions are more concentrated in fewer locations, which suggests it might focus on more predictable, routine movements but could **miss out on less frequently visited POIs**.
> * In summary, LoTNext not only provides a **better balance** between exploring different areas and accurately predicting the next POI but also successfully predicts the actual next POI, compared with Graph-Flashback.
>
> > Q3. A more detailed literature review is needed (at least in the appendix) so that the novelty of the method could be better evaluated.
>
> **A3:** Thanks for your helpful advice. We will include a more detailed literature review in the final version, covering topics such as LLM-based human mobility prediction and other long-tailed learning methods. In addition, we would like to conclude our novelty as below:
> * Due to the **input** of the human mobility prediction task being **not individual samples** but **sequences**, many long-tailed learning methods from CV and recommendation field are **hard to deploy** in the human mobility prediction task.
> * In the human mobility prediction community, the current focus is on improving model prediction accuracy from the perspective of graph optimization learning, completely **overlooking the long-tail feature of human check-in datasets**.
> * **Our study is the first to propose a general framework for human mobility prediction under the long-tail problem**.
> * **LLM-based human mobility prediction** is still in its early stages and holds significant potential. We believe that exploring and addressing the **long-tail problem based on the LLM framework** is a very interesting direction for future research.
>
> > Q4. The comparison methods used are somewhat outdated. Why didn't you use the latest methods, such as TPG or LLM-Move, for comparison?
>
> **A4:** Thank you for your detailed comments and valuable paper recommendations. **LLM-Move [6]** is based on large language models (LLMs) for human mobility prediction tasks. However, **LLM-Move [6]** requires category as additional semantic information, which is not applicable in our dataset.
>
> Therefore, we add **AGRAN [3]** suggested by reviewer 7ppu and **TPG [5]** as our new baselines, please refer to the **"global" response** (please see above) for a detailed experiment results explanation.
>
> **Questions:**
> > Q5. Why does cutting off the long tail in the user-POI graph for noise reduction, followed by using loss to reintroduce the long tail, theoretically improve long tail prediction?
>
> **A5:** Thank you for your questions. We do not reintroduce the long tail through the loss function; instead, we ensure that the model does not overly focus on head data.
> * First, the long-tailed loss adjustment module increases the attention to tail data via logit score adjustment.
> * Next, the sample weight adjustment module calculates the vector magnitude of all samples and compares them to the geometric mean magnitude to identify those samples that contribute significantly to the model.
> * This approach balances the learning of features from both head and tail data.
>
> > Q6. In Figure 4, how should we interpret "four least frequently occurring pois"? Does it refer to only the four POIs with the lowest frequencies? I guess many POIs only appear once.
>
> **A6:** Thank you for pointing out this confusing problem. We apologize for the unclear explanation here. Specifically, we divide the least frequently occurring POIs into four groups using the intervals [1,10), [10,20), [20,30), and [30,40). Then, we perform t-SNE visualization on these groups.
>
> *If our responses could address your concerns, please kindly consider raising the score.*

---

> > ### Author Response · Authors · 2024-08-11
> >
> > Thank you for your comments. To ensure a thorough evaluation, I would appreciate it if you could review the rebuttal and provide any further feedback. Your input is greatly valued.

---

> ### Comment · Area_Chair_odhj · 2024-08-13
>
> Dear reviewer,
>
> Could you please read the rebuttal and engage the discussion with the authors ASAP?
>
> AC

---

> > ### Comment · Reviewer_Et5T · 2024-08-14
> >
> > Thanks for your detailed response. Most of my concerns have been addressed. I will raise the score.

---

> > > ### Author Response · Authors · 2024-08-14
> > >
> > > Thank you for your detailed review and for acknowledging the responses. If you are convenient, you may update your score by editing your original review. I truly appreciate your time and consideration.

---

> ### Author Response · Authors · 2024-08-14
> **Update Score Reminder**
>
> The discussion period will end within an hour, the comment shows you would **increase the score** (perhaps you just forgot to do it...), could you please **kindly do it**?

---

### Official Review · Reviewer_7ppu · 2024-07-17

**Soundness:** 2
**Presentation:** 3
**Contribution:** 3
**Rating:** 5
**Confidence:** 4

**Summary:**

This study proposes the Long-Tail Adjusted Next Point-of-Interest Prediction (LoTNext) framework. By combining a Long-Tailed Graph Adjustment module and a Long-Tailed Loss Adjustment module, it reduces the impact of long-tailed nodes in the user-POI interaction graph and adjusts loss through logit score and sample weight adjustment strategies. Experimental results show that LoTNext outperforms several existing methods on two real-world datasets.

**Strengths:**

1. The structure and organization of this paper are well-designed, and the writing is clear and easy to comprehend.
2. This paper investigates the long-tail problem by proposing a general framework for next POI recommendation, filling the gap in addressing the long-tail issue in POI recommendation. This work is meaningful and valuable.
3. To enhance the readability of the paper, the authors provide detailed results analysis, parameter settings, and the motivation behind the design of each module in the appendix.

**Weaknesses:**

1.  In the related work section, the authors review common methods for addressing the long-tail problem in recommendation systems. Since this paper focuses on addressing the long-tail problem, adding several baselines that tackle the long-tail issue in recommendation systems (e,g,, [1]) would better demonstrate the effectiveness of the proposed method.
2. The novelty of this paper is not very strong. The long tail effect of check-in data, such as the POI frequency distributions, has been studied before.
3. Additional comparative analyses should be included to illustrate the shortcomings of baselines in handling the long-tail issue. For instance, comparing the proposed model's performance with all baselines (not just Graph-Flashback) on long-tail POIs would better demonstrate its effectiveness in addressing the long-tail problem.
4. The experimental results are not convincing enough, as the compared methods are not the SOTA method. More recent baselines should be compared (e.g., [2-4]).

[1] Meta graph learning for long-tail recommendation, SIGKDD, 2023.
[2] EEDN: Enhanced Encoder-Decoder Network with Local and Global Context Learning for POI Recommendation， SIGIR-23
[3] Adaptive Graph Representation Learning for Next POI Recommendation， SIGIR-23
[4] Spatio-Temporal Hypergraph Learning for Next POI Recommendation， SIGIR-23

**Questions:**

1. How do the authors balance the performance between head POIs and long-tail POIs? In other words, how do they enhance the performance of long-tail POIs without compromising the performance of head POIs? As shown in Figure 3(c), the proportion of predicted long-tail POIs is relatively high. Does this affect the prediction of head POIs?
2. In the experimental section, it would be beneficial to show the performance of the proposed method and the baselines on both head POIs and long tail POIs to enhance the persuasiveness of the conclusions.

**Limitations:**

The authors have already pointed out a limitation of this method: LoTNext relies on extensive user trajectory data, which, if deployed by certain institutions or companies, may pose a potential risk of privacy breaches, potentially leading to negative social impacts.

---

> ### Author Rebuttal · Authors · 2024-08-06
>
> # Response to Reviewer 7ppu:
> **Weaknesses:**
> > Q1. In the related work section, the authors review common methods for addressing the long-tail problem in recommendation systems. Since this paper focuses on addressing the long-tail problem, adding several baselines that tackle the long-tail issue in recommendation systems (e,g, [1]) would better demonstrate the effectiveness of the proposed method.
>
> **A1:** Thank you for your suggestions and valuable references. Compared with MGL [1], our task has several difference:
> 1. MGL [1] primarily addresses the click problem under long-tail problem, i.e., whether or not to interact with an item, without considering the temporal aspect. In contrast, our task in human mobility prediction inherently involves spatial-temporal features, such as check-in times and POI geographic locations.
> 2. The input of MGL is the user's feedback on item ratings, clicks, etc., without the temporal features. However, our model input is a trajectory sequence with spatial-temporal features, hence MGL is difficult to deploy under our datasets.
>
> Due to time and computation resource constraints, we add the AGRAN [3] model as our new baseline. Additionally, we also include the TPG [5] as a new baseline suggested by reviewer Et5T. The specific results are shown in **"global" response** (please see above).
>
> > Q2. The novelty of this paper is not very strong. The long tail effect of check-in data, such as the POI frequency distributions, has been studied before.
>
> **A2:** Thanks for your insightful comments. Different with previous work, our paper has several novel points:
> 1. We are the **first work** to propose a solution specifically for the **long-tail problem in the human mobility prediction field**.
> 2. Our approach includes a graph adjustment module, a loss adjustment module, and a time prediction auxiliary module. Through these novel modules, we significantly enhance the model's performance in predicting long-tail data (from Table 1 and Figure 3) .
> 3. Previous studies have only treated check-in frequency as a data feature of POIs, which is fundamentally different from our approach.
>
> > Q3. Additional comparative analyses should be included to illustrate the shortcomings of baselines in handling the long-tail issue. For instance, comparing the proposed model's performance with all baselines (not just Graph-Flashback) on long-tail POIs would better demonstrate its effectiveness in addressing the long-tail problem.
>
> **A3:** Thanks for your detailed comments. Due to time constraints, we apologize for not being able to compare the performance of all baselines on tail data. Given our inclusion of the graph adjustment module, we focused on graph-based baselines. As shown in Table 1, Graph-Flashback and SNPM are the best-performing graph-based baselines. Following your suggestion, we included SNPM for comparison. The detailed results are shown in Figure I attached in the **"global" response** PDF file (please see above).
> * From Figure I, we can find on both the Gowalla and Foursquare datasets, LoTNext outperforms other baselines in predicting tail data, particularly evident in the MRR metric.
> * Moreover, we do not compromise the prediction of head data, as LoTNext also shows superior performance compared to the other two baselines in head data prediction.
>
> > Q4. The experimental results are not convincing enough, as the compared methods are not the SOTA method. More recent baselines should be compared (e.g., [2-4]).
>
> **A4:** Thank you for your suggestions. Please refer to the answer of Q1 in detail.
>
> **Questions:**
> > Q5. How do the authors balance the performance between head POIs and long-tail POIs? In other words, how do they enhance the performance of long-tail POIs without compromising the performance of head POIs? As shown in Figure 3(c), the proportion of predicted long-tail POIs is relatively high. Does this affect the prediction of head POIs?
>
> **A5:** Thank you for your valuable questions. In our loss adjustment module, the logit score adjustment module prevents the model from overly focusing on head data, while the sample weight adjustment module balances head and tail data.
> * First, the logit score adjustment module enhances the weights of tail samples by POI visit frequency.
> * Next, the sample weight adjustment module calculates the vector magnitude of each sample, and comparing it to a baseline magnitude, we penalize samples that are less beneficial for the model's predictions.
> * In summary, regardless of whether the data is head or tail, if the model determines that a sample contributes significantly overall, we increase its weight in the loss to ensure the model treats head and tail samples equitably.
> * In addition, as you mentioned, according to Figure 3(c), LoTNext predicts a higher proportion of tail samples. However, this does not affect the model's overall performance. Figures 3(a) and (b) and Figures Ⅰ(a), (b), (c), and (d) in the **"global" response** PDF file (see above) demonstrate that our LoTNext outperforms other baselines in predicting both head and tail data.
>
> > Q6. In the experimental section, it would be beneficial to show the performance of the proposed method and the baselines on both head POIs and long tail POIs to enhance the persuasiveness of the conclusions.
>
> **A6:** Thank you for your suggestions. Figure 3(a) and (b) include the relevant experiments. Additionally, in response to your comments on Q3 and Q5, we have incorporated other baselines for comparison. Please refer to our responses to Q3 and Q5 for detailed information.
>
> *If you are satisfied with our responses, we would be grateful if you could kindly check the positive remarks of other reviewers and consider raising the score.*

---

> > ### Author Response · Authors · 2024-08-11
> >
> > Thank you for your initial feedback. If you have a moment, I would greatly appreciate any further thoughts on the points addressed in my rebuttal. Your insights are crucial for improving the quality of the work.

---

> > ### Comment · Reviewer_7ppu · 2024-08-13
> > **Rebuttal has been checked.**
> >
> > Thanks for the authors' rebuttal. I have checked the responses and other reviewers' reviews. Some concerns have been addressed. Hence, I would increase my scores.

---

> > > ### Author Response · Authors · 2024-08-13
> > >
> > > Thank you for taking the time to review the rebuttal and the other reviewers' comments. I appreciate your willingness to update your evaluation based on our provided rebuttal.

---

### Author Rebuttal · Authors · 2024-08-06

# Response to All Reviewers:
We thank the reviewer for the very valuable, detailed and constructive feedback on our work. We especially thank the positive words:

* work is meaningful and valuable, worthwhile issue to study (Reviewer #7ppu & #Et5T)
* filling the gap in addressing the long-tail issue (Reviewer #7ppu & #Et5T)
* well-written, easy to comprehend and thorough explanation of the methodology (Reviewer #7ppu & #mZab & #JhGF)
* motivation explanation behind the design of each module (Reviewer #7ppu &#JhGF)
* detailed results analysis and superior performance across multiple metrics (Reviewer #7ppu & #mZab)
* significant contribution to the field of human mobility prediction (Reviewer #mZab)
* the reproducibility of this paper is good (Reviewer #mZab & #JhGF)

Due to word limitation, we conclude a similar question and answer it in response to your valuable comments. Thank you!

> Q1 Need to add more baselines, metrics and datasets. (Reviewer #7ppu & #Et5T)

**A1:**
Based on the feedback from reviewers #7ppu and #Et5T, we have summarized six highly valuable and relevant papers, each with different focuses:

1. **MGL [1]** focuses on addressing the long-tail problem in the recommendation domain. The input of MGL is the **user's feedback** on item ratings, clicks, etc., without the temporal features. However, our model input is a **trajectory sequence** with spatial-temporal features, hence MGL is difficult to deploy under our datasets.

2. **EEDN [2]** proposes an enhanced network to tackle implicit feedback and cold-start issues in POI recommendation by leveraging latent interactions between users and POIs. However, its **input trajectories lack specific temporal factors**, making this work more akin to the sequential recommendation and slightly different from our task definition.

3. **AGRAN [3]** proposes an adaptive graph representation method, which explores the utilization of graph structure learning to replace static graphs.

4. **STHGCN [4]** proposes to capture the higher-order information including user trajectories and the collaborative relations among trajectories by hypergraph. However, the construction of hypergraph requires **categories semantic information**, which is not applicable in our dataset.

5. **TPG [5]** proposes a framework that integrate temporal prompts and geography-aware strategies, overcoming encoding limitations of traditional methods.

6. **LLM-Move [6]** is based on large language models (LLMs) for human mobility prediction tasks. However, LLM-Move requires additional **category semantic information** as input, which is not applicable in our dataset.

In summary, we add AGRAN [3] and TPG [5] as new baselines to further validate the performance of our model. Additionally, we add Normalized Discounted Cumulative Gain (NDCG) as a new metric suggested by reviewer #Et5T (since the results of NDCG@1 and Acc@1 are same, we ignore it here). For better comparison, we only excerpt the best-performing baselines from the original paper. The detailed results are as follows:

Table Ⅰ. The performance comparison on Gowalla dataset.
|Method/Metric|Acc@1|Acc@5|Acc@10|MRR|NDCG@5|NDCG@10|
|-|-|-|-|-|-|-|
|Graph-Flashback|0.1495|0.3399|0.4242|0.2401|0.2497|0.2766|
|SNPM|0.1593|0.3514|0.4346|0.2505|0.2600|0.2872|
|AGRAN (new)|0.1005|0.2456|0.3154|0.1731|0.1764|0.1990|
|TPG (new)|0.1400|0.3071|0.3611|0.1948|0.2059|0.2374|
|**LoTNext**|**0.1668**|**0.3605**|**0.4429**|**0.2591**|**0.2686**|**0.2953**|

Table Ⅱ. The performance comparison on Foursquare dataset.
|Method/Metric|Acc@1|Acc@5|Acc@10|MRR| NDCG@5|NDCG@10|
|-|-|-|-|-|-|-|
|Graph-Flashback|0.2786|0.5733|0.6501|0.4109|0.4411|0.4661|
|SNPM|0.2899|0.5967|0.6763|0.4278|0.4480|0.4757|
|AGRAN (new)|0.1575|0.3736|0.4676|0.2600|0.2703|0.3008|
|TPG (new)|0.2321|0.4631|0.5493|0.3775|0.3891|0.4106|
|**LoTNext**|**0.3155**|**0.6059**|**0.6812**|**0.4469**|**0.4753**|**0.5001**|

* **AGRAN [3]** perform lower than the graph-based baselines such as **Graph-Flashback** and **SNPM**.  However, the deployed datasets from **AGRAN [3]** are based on Foursquare-Singapore and Gowalla-Nevada, which are **city-level** making adaptive graph learning more suitable.
* Our datasets are **global-level**, as shown in the heatmap from Figure 6 in the appendix, which are significantly larger than city-level datasets. Therefore, it is challenging to adaptative build and learn embeddings from graphs on global-level datasets without relying on prior knowledge. These factors contribute to the lower performance of **AGRAN [3]** under our global human check-in datasets.

* **TPG [5]** performs better than **AGRAN [5]** and is very close to **STAN**. However, it requires knowing the exact time of the next location visit during prediction.  **LoTNext** does not need future moments as prompts and performs better.
* **TPG [5]** explicitly models the geography of longitude and latitude, which poses a potential risk of user privacy leakage. **LoTNext** calculates spatial contextual attention based on geographic distance differences, which has a lower risk of user privacy leakage.

**References**

[1] Wei, C., et al. Meta graph learning for long-tail recommendation. KDD 2023.

[2] Wang, X., et al. Eedn: Enhanced encoder-decoder network with local and global context learning for poi recommendation. SIGIR 2023.

[3] Wang, Z., et al. Adaptive Graph Representation Learning for Next POI Recommendation. SIGIR 2023.

[4] Yan, X., et al. Spatio-temporal hypergraph learning for next POI recommendation. SIGIR 2023.

[5] Luo, Y., et al. Timestamps as Prompts for Geography-Aware Location Recommendation. CIKM 2023.

[6] Feng, S., et al. Where to move next: Zero-shot generalization of llms for next poi recommendation. IEEE CAI 2024.

---

### Author Response · Authors · 2024-08-14
**Summary of Reviews and Rebuttals**

**Table Reviewer Rating Summarization.**

| Score/Reviewer | Reviewer 7ppu | Reviewer Et5T                                         | Reviewer mZab | Reviewer JhGF | Average    |
| -------------- | ------------- | ----------------------------------------------------- | ------------- | ------------- | --- |
| Rating         | 5             | 5 (said will **increase the score** from 5 to higher) | 6             | 7             |  5.75 (will be **higher** due to **unfinished score update**)   |
| Confidence    | 4             | 5                                                     | 3             | 5             |  4.25   |

From the table, it is clear that **all reviewers** have shown a **positive attitude** towards our paper and have **high confidence** in it.

Our work is the **first work** to address **the long-tail problem in human mobility modeling tasks**. We believe it did make contributions to the machine learning for social sciences area. Hope we could get your kind support.

---

### Decision · Program_Chairs · 2024-09-25

**Decision:**

Accept (poster)

**Comment:**

The paper introduces the Long-Tail Adjusted Next POI Prediction (LoTNext) framework to tackle the long-tail problem in next POI prediction. LoTNext features a Long-Tailed Graph Adjustment module to minimize noise and mitigate the influence of long-tailed nodes in the user-POI interaction graph, as well as a Long-Tailed Loss Adjustment module to balance the loss between head and tail POIs. Experimental results demonstrate that LoTNext outperforms several existing methods on two real-world datasets. After the rebuttal, the overall rating of the paper turned positive, indicating a leaning towards acceptance. However, some of the authors' feedback appears to be either unreasonable or inadequately addresses the reviewers' concerns (including baseline selection, scalability and the existing work about handling long-tailed issue ).